# Physics-Informed Distillation of Diffusion Models for PDE-Constrained Generation

## Abstract

Modeling physical systems in a generative manner offers several advantages, including the ability to handle partial observations, generate diverse solutions, and address both forward and inverse problems. Recently, diffusion models have gained increasing attention in the modeling of physical systems, particularly those governed by partial differential equations (PDEs). However, diffusion models only access noisy data $x_t$ at intermediate steps, making it infeasible to directly enforce constraints on the clean sample $x_0$ at each noisy level. As a workaround, constraints are typically applied to the *expectation* of clean samples $\mathbb{E}[x_0|x_t]$, which is estimated using the learned score network. However, imposing PDE constraints on the expectation does not strictly represent the one on the true clean data, known as *Jensen's Gap*. This gap creates a trade-off: enforcing PDE constraints may come at the cost of reduced accuracy in generative modeling. To address this, we propose a *simple* yet *effective* post-hoc distillation approach, where PDE constraints are not injected directly into the diffusion process, but instead enforced during a post-hoc distillation stage. We term our method as **P**hysics-**I**nformed **D**istillation of **D**iffusion **M**odels (PIDDM). This distillation not only facilitates single-step generation with improved PDE satisfaction, but also support both forward and inverse problem solving and reconstruction from randomly partial observation. Extensive experiments across various PDE benchmarks demonstrate that PIDDM significantly *both* improves PDE satisfaction and generative modeling over several recent and competitive baselines, such as PIDM (3), DiffusionPDE (22), and ECI-sampling (8), while achieving lower computational overhead and avoiding extensive hyperparameter tuning. Our approach can shed light on more efficient and effective strategies for incorporating physical constraints into diffusion models.

## 1 Introduction

Solving partial differential equations (PDEs) underpins innumerable applications in physics, biology, and engineering, spanning fluid flow (11), heat transfer (25), elasticity (64), electromagnetism (26), and chemical diffusion (10). Classical discretisation schemes such as finite-difference (57), and finite-element methods (33) provide reliable solutions, but their computational cost grows sharply with mesh resolution, dimensionality, and parameter sweeps, limiting their practicality for large-scale or real-time simulations (24). This bottleneck has fuelled a surge of learning-based solvers that approximate or accelerate PDE solutions, from early physics-informed neural networks (PINNs) (51) to modern operator-learning frameworks such as DeepONet (43), Fourier Neural Operators (35) and Physics-informed Neural Operator (36), offering faster inference, uncertainty quantification, and seamless integration into inverse or data-driven tasks.

Among these learning based solvers, diffusion models (20; 61) provide a promising framework for generative modeling of physical systems. For PDEs, a diffusion model can learn the joint distribution of solution and coefficient fields, $x_0 = (u, a)$, from data, where $a$ denotes input parameters that satisfy the boundary operator $\mathcal{B}$ (for example, material properties or initial conditions) and $u$ is the corresponding solution that satisfies the PDE operator $\mathcal{F}$. After training, the model can sample $(u, a)$ from this learned distribution, enabling forward simulation (sample $u$ given $a$), inverse recovery (sample $a$ given $u$), and conditional reconstruction (complete missing components of $u$ or $a$) *within a single framework, which prior non-diffusion approaches (35; 43; 51) do not provide.* However, while

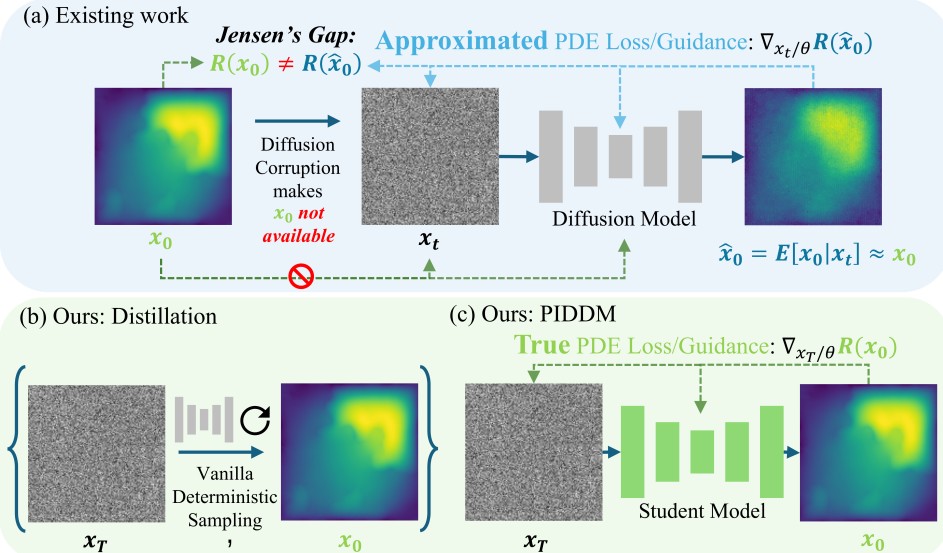

Figure 1: **Illustration of physics-constrained diffusion generation and our proposed framework.** **(a)** Existing methods (22; 8; 3; 27) impose PDE losses or guidance on the posterior mean $\mathbb{E}[x_0|x_t]$ in diffusion training and sampling, introducing Jensen's Gap. **(b)** We propose to train and sample diffusion model using vanilla methods to generate a noise-image data paired dataset for distillation. **(c)** Our proposed framework distills the teacher diffusion model and directly enforces physical constraints on the final generated sample $x_0$, avoiding Jensen's Gap .

diffusion models perform well under soft, high level constraints (53; 16; 21; 9; 4), PDE applications often require strict, low level constraints dictated by $\mathcal{F}$ and $\mathcal{B}$.

Enforcing such PDE constraints within diffusion models is nontrivial. A core difficulty is that, at individual noise level $t$, diffusion models operate on noisy variables $x_t$ rather than the clean physical field $x_0$, where constraints such as $\mathcal{F}[x_0] = 0$ are defined. To address this, one option is to reconstruct $x_0$ by running the full deterministic sampling trajectory, but this is computationally expensive since it requires many forward passes, and enforcing constraints through backpropagation often causes gradient issues. (3). A more common alternative is to approximate $x_0$ with the posterior mean $\mathbb{E}[x_0|x_t]$, which can be efficiently computed via Tweedie's formula (3; 22; 8; 27; 29; 68; 76; 71; 59; 62; 23) (see right part of Fig. 1 (a)). However, this introduces a theoretical inconsistency: enforcing constraints on the posterior mean, $\mathcal{F}[\mathbb{E}[x_0|x_t]]$, is not equivalent to enforcing the expected constraint, $\mathbb{E}[\mathcal{F}[x_0]|x_t]$, due to Jensen's inequality. This mismatch, known as the *Jensen's Gap* (3), can lead to degraded physical fidelity.

**Contributions.** We propose a simple yet effective framework that enforces PDE constraints in diffusion models via post-hoc distillation, enabling reliable and efficient generation under physical laws. As shown in Fig. 1 (c), our method sidesteps the limitations of existing constraint-guided diffusion-based approaches by decoupling physics enforcement from the diffusion trajectory. Our main contributions are:

- **Empirical confirmation of Jensen's Gap:** We provide the first explicit empirical demonstration and quantitative analysis of the *Jensen's Gap*, a fundamental discrepancy that arises when PDE constraints are imposed on the posterior mean $\mathbb{E}[x_0|x_t]$, rather than the final clean sample $x_0$.
- **Theoretically sound:** Our method bypasses the Jensen's Gap by enforcing PDE constraints on the final generated samples via distillation. Unlike posterior-mean-based methods that trade distributional fidelity for constraint satisfaction, our method achieves both physical accuracy and generative fidelity without extensive hyperparameter tuning.
- **Versatile and efficient inference:** The distilled student model preserves the full generative capabilities of the teacher, supporting physical simulation, reconstruction, and unified forward and inverse PDE solving within *a single model*, while enabling one step generation for fast inference. Experiments across diverse PDEs show that PIDDM surpasses posterior-mean-based methods (22; 8; 3; 27) in both generation quality and constraint satisfaction.

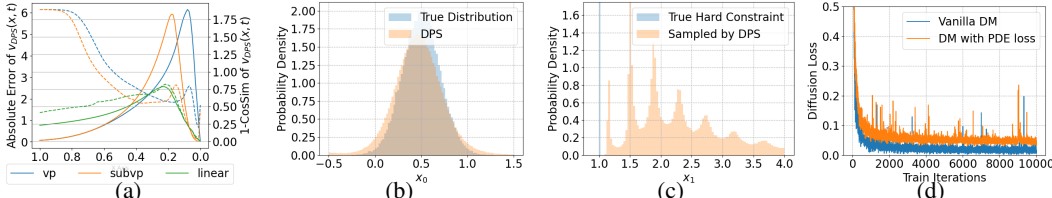

(a)  (b)  (c)  (d)

Figure 2: **Empirical illustration of the Jensen's Gap in physics-constrained diffusion models. (a)** Absolute velocity error and angular discrepancy $(1 - \cos(\theta))$ between Diffusion Posterior Sampling (DPS) and the ground-truth conditional ODE velocity on the MoG dataset. **(b)** and **(c)** Histograms comparing the first (unconstrained) and second (hard-constrained) dimensions of DPS-sampled MoG data against the ground truth MoG. **(d)** Training-time manifestation: diffusion loss comparison between vanilla training and PIDM on a Stokes Problem dataset.

## 2 Problem Setup: Jensen's Gap in Diffusion Model with PDE Constraints

In scientific machine learning, there exist many *hard* and *low-level* constraints that are mathematically strict and non-negotiable (34; 19; 48; 54). In this section, we will discuss how existing works impose these constraints in diffusion-generated data, and the Jensen's Gap (17; 3; 22) it introduces.

### 2.1 Preliminaries on Physics constraints

Physics constraints are typically expressed as a *partial differential equation (PDE)* $\mathcal{F}$ defined over a solution domain $\Omega \subset \mathbb{R}^d$, together with a *boundary condition* operator $\mathcal{B}$ defined on the coefficient domain $\Omega'$:

$$\mathcal{F}[\mathbf{u}(\boldsymbol{\xi})] = 0 \text{ for } \boldsymbol{\xi} \in \Omega, \quad \mathcal{B}[\mathbf{a}(\boldsymbol{\xi}')] = 0 \text{ for } \boldsymbol{\xi}' \in \Omega'. \tag{1}$$

In practice, the domain $\Omega$ and $\Omega'$ is discretized into a uniform grid, typically of size $H \times W$, and the field $\boldsymbol{u}$ and $\boldsymbol{a}$ are evaluated at those grid points to produce the observed data $\boldsymbol{x}_0 = (\boldsymbol{u}, \boldsymbol{a})$, where diffusion models are trained to learn the joint distribution $p(\boldsymbol{x}_0) = p((\boldsymbol{u}, \boldsymbol{a}))$. While PINNs (51) model the mapping $\boldsymbol{\xi}, \boldsymbol{\xi}' \mapsto (\mathbf{u}(\boldsymbol{\xi}), \mathbf{a}(\boldsymbol{\xi}'))$ with differentiable neural networks to enable automatic differentiation (50; 1), grid-based approaches commonly approximate the differential operators in $\mathcal{F}$ via finite difference methods (57; 33). To quantify the extent to which a generated sample $\boldsymbol{x}_0$ violates the physical constraints, the *physics residual error* in often defined by:

$$\mathcal{R}(\boldsymbol{x}_0) = \mathcal{R}((\boldsymbol{u}, \boldsymbol{a})) := \left[\mathcal{F}[\boldsymbol{u}], \mathcal{B}[\boldsymbol{a}]\right]^\top \tag{2}$$

Here, $\mathcal{R}(\boldsymbol{x})$ measures the discrepancy between the sample $\boldsymbol{x}$ and the expected PDE $\mathcal{F}$ and boundary conditions $\mathcal{B}$. The *physics residual loss* is often defined by the squared norm of this physics residual error, i.e., $\|\mathcal{R}(\boldsymbol{x})\|^2$.

### 2.2 Imposing PDE constraints in Diffusion Models

The physical constraints $\mathcal{R}$ are often defined on the clean field $\boldsymbol{x}_0$, while during training or sampling of the diffusion model, the model only observes the noisy state $\boldsymbol{x}_t$. Therefore, it is intractable to make direct optimization or controlled generation based on the physical residual loss $\mathcal{R}(\boldsymbol{x}_0)$. A practical workaround, therefore, is to evaluate the constraint on an estimate of $\boldsymbol{x}_0$ from $\boldsymbol{x}_t$, and a common choice is to use the estimated *posterior mean*: $\mathbb{E}[\boldsymbol{x}_0 \mid \boldsymbol{x}_t]$ based on the score network in diffusion model (3; 22). As a simplified example, consider the forward process defined as $\boldsymbol{x}_t = \boldsymbol{x}_0 + \sigma_t \boldsymbol{\epsilon}$, where $\sigma_t$ denotes the noise level at time $t$ and $\boldsymbol{\epsilon} \sim \mathcal{N}(0, \mathbf{I})$ is standard Gaussian noise. Then, the posterior mean can be efficiently estimated via Tweedie's formula:

$$\hat{\boldsymbol{x}}_\theta(\boldsymbol{x}_t, t) := \boldsymbol{x}_t + \sigma_t^2 s_\theta(\boldsymbol{x}_t, t) \approx \boldsymbol{x}_t + \sigma_t^2 \nabla \log p(\boldsymbol{x}_t) \approx \mathbb{E}[\boldsymbol{x}_0 \mid \boldsymbol{x}_t], \tag{3}$$

where $s_\theta$ is a learned score function approximating the gradient of the log-density (see Appendix B.2 for the derivations for the general diffusion process). Leveraging this approximation, several existing works incorporate PDE constraints by evaluating the PDE residual operator $\mathcal{R}$ on $\hat{\boldsymbol{x}}_\theta(\boldsymbol{x}_t, t)$. For instance, **PIDM** (3) integrates PDE constraints into diffusion model *at training time* by augmenting the standard diffusion objective with an additional PDE residual loss $\mathcal{R}(\hat{\boldsymbol{x}}_\theta(\boldsymbol{x}_t, t))$. Similarly, *at*

*inference time*, **DiffusionPDE** (22) and **CoCoGen** (27) employ diffusion posterior sampling (DPS) (9), guiding each intermediate sample $\boldsymbol{x}_t$ using the gradient $\nabla_{\boldsymbol{x}} \mathcal{R}(\hat{\boldsymbol{x}}_\theta(\boldsymbol{x}_t, t))$. On the other hand; **ECI-sampling** (8) directly projects hard constraints onto the posterior mean at each DDIM step using a correction operator (more detailed discussion on their implementations can be found in Appendix D.4). Beyond PDE applications, constrained diffusion models for image inverse problems also rely on posterior-mean approximations for true posterior estimation. Representative examples include DDRM (29), DDNM (68), LGD (59), DPG (62), SCG (23), DCDPM (13), mid-point guidance (47), DiffPIR (76), and DAPS (71) (see Appendix A.2 for details). While these pioneering methods have been demonstrated to be effective in enforcing PDE constraints within diffusion models, they still suffer a theoretical inconsistency: PDE constraints are enforced on the posterior mean approximation $\mathbb{E}[\boldsymbol{x}_0|\boldsymbol{x}_t]$, which is not equivalent to the constraints on the true generated data $\boldsymbol{x}_0$ due to Jensen's inequality:

$$\mathcal{R}(\mathbb{E}[\boldsymbol{x}_0|\boldsymbol{x}_t]) \neq \mathbb{E}[\mathcal{R}(\boldsymbol{x}_0)|\boldsymbol{x}_t]. \tag{4}$$

This discrepancy is commonly referred to as the *Jensen's Gap* (3; 22; 17). To mitigate this issue, PIDM and DiffusionPDE heuristically down-weight PDE constraints at early denoising steps (large $t$) in training and sampling, respectively, where Jensen's Gap is pronounced, and emphasize them near $t \rightarrow 0$, where the posterior mean approximation improves. ECI-sampling introduces stochastic resampling steps (66) to project theoretically inconsistent intermediate samples back toward their correct distribution. Many other methods for image inverse problems also provide partial improvements to reduce approximation error (59; 62; 23; 13; 47) (Appendix A.2). However, they merely mitigate the Jensen's Gap rather than fundamentally resolving it.

### 2.3 DEMONSTRATION OF THE JENSEN'S GAP

To better illustrate the presence of Jensen's gap and its negative effect, we conduct experiments on two synthetic datasets: a Mixture-of-Gaussians (MoG) dataset and a Stokes Problem dataset.

**Sampling-time Jensen's Gap.** We demonstrate the sampling-time Jensen's Gap using the Mixture-of-Gaussians (MoG) dataset, where the score function is analytically tractable, allowing us to isolate the effect of the diffusion process without interference from training error. The MoG is constructed in 2D: the first dimension follows a bimodal Gaussian distribution, while the second dimension encodes a discrete latent variable that serves as a hard constraint. Concretely, the joint distribution is defined as a mixture of two Gaussians, each supported on a distinct horizontal line:

$$p(x_0) = 0.5 \cdot \mathcal{N}(x_1; -1, \sigma^2) \cdot \delta(x_2 + 1) + 0.5 \cdot \mathcal{N}(x_1; +1, \sigma^2) \cdot \delta(x_2 - 1), \tag{5}$$

where $\delta(\cdot)$ denotes the Dirac delta function and $\sigma = 0.2$. To examine the impact of Jensen's Gap during sampling, we compare Diffusion Posterior Sampling (DPS) (9) which uses a latent code to guide the generation, with the ground-truth conditional ODE trajectory derived analytically. We evaluate three representative diffusion processes: Variance-Preserving (VP) (20), Sub-VP (61), and Linear (38), and compare their velocity field during the inference for characterizing Jensen's Gap. To quantify amplitude errors, we compute the mean absolute error (MAE) and angular error between the DPS-predicted velocity field $v_{\text{DPS}}(x, t)$ and the ground-truth velocity $v_{\text{GT}}(x, t)$: We observe that both of these errors of DPS are significantly elevated at intermediate timesteps and only diminish as $t \rightarrow 0$, as shown in Fig.2a. Although DPS achieves accurate sampling in the unconstrained dimension (Fig.2b), it fails to respect the hard constraint in the constrained dimension (Fig. 2c).

**Training-time Jensen's Gap.** We examine the Jensen's Gap during training using the synthetic Stokes dataset with target distribution $p_{\text{Stokes}}$. The diffusion model $\boldsymbol{v}_\theta$ adopts a Fourier Neural Operator (FNO) (35) architecture and follows a standard linear noise schedule (37; 39; 40). Dataset and training details are given in Appendices C and D. We take PIDM (3) as a representative method, which augments the diffusion loss with a PDE residual term $\mathcal{R}(\hat{\boldsymbol{x}}_\theta(\boldsymbol{x}_t, t))$, and compare its performance with standard diffusion training. To assess generative quality, we monitor the diffusion loss, which theoretically corresponds to the evidence lower bound (ELBO) (20; 32; 12; 52; 15). The comparison results are shown in Fig. 2d, revealing a significant increase in diffusion loss when the PDE residual loss is incorporated. This suggests that the PDE residual loss does not help better shape the data distribution that satisfies the PDE constraints. This observation also corroborates findings from PIDM (3), which identified that residual supervision on the posterior mean can create "a conflicting objective between the data and residual loss", where the data loss represents the original diffusion training objective. These results provide further evidence for the existence of the Jensen's Gap in training, as enforcing constraints on $\mathbb{E}[\boldsymbol{x}_0|\boldsymbol{x}_t]$ may interfere with maximizing the likelihood of the true data distribution.

---

**Algorithm 1** PIDDM Training: Physics-Informed Distillation.

---

**Require:** Teacher Model $v_\theta(x, t)$, Student Model $d_{\theta'}$, Batch Size $B$, Steps $N_s$, Step Size $\mathrm{d}t=1/N_s$, Physics Residual Error $\mathcal{R}$, Loss Weight $\lambda_{\text{train}}$, Learning Rate $\eta_{\text{train}}$

1: **repeat**
2:      Sample $\epsilon_{1:B} \overset{\text{i.i.d.}}{\sim} \mathcal{N}(\mathbf{0}, \mathbf{I})$;    $x_T \leftarrow \epsilon_{1:B}$
3:      **For** $t = T-\mathrm{d}t, \ldots, 0$    $x_t \leftarrow x_{t+\mathrm{d}t} - v_\theta(x_{t+\mathrm{d}t}, t+\mathrm{d}t) \, \mathrm{d}t$        ▷ **Sampling Phase**
4:      $x_{\text{pred}} \leftarrow d_{\theta'}(\epsilon_{1:B})$
5:      $\mathcal{L} \leftarrow \frac{1}{B}(\|x_{\text{pred}} - x_0\|^2 + \lambda_{\text{train}}\|\mathcal{R}(x_{\text{pred}})\|^2)$        ▷ **Distillation Phase**
6:      $\theta' \leftarrow \theta' - \eta_{\text{train}}\nabla_{\theta'}\mathcal{L}$
7: **until** Converged

---

**Algorithm 2** PIDDM Inference: Physics Data Simulation

---

1: **Input** Student Model $d_{\theta'}$, Physics Residual Error $\mathcal{R}$, Refinement Step Number $N_f$, Refinement Step Size $\eta_{\text{ref}}$, Latent Noise $\epsilon \sim \mathcal{N}(\mathbf{0}, \mathbf{I})$.
2: **For** $i = 1, \ldots, N_f : \epsilon \leftarrow \epsilon - \eta_{\text{ref}}\nabla_\epsilon\|\mathcal{R}(d_{\theta'}(\epsilon))\|^2$        ▷ PDE refinement step (optional).
3: **Output** $d_{\theta'}(\epsilon)$

---

**Algorithm 3** PIDDM Inference for Forward/Inverse/Reconstruction

---

1: **Input** Student Model $d_{\theta'}$, Physics Residual Error $\mathcal{R}$, Optimization Iteration $N_o$, Step Size $\eta_{\text{infer}}$, Observation $x'$, Observation Mask $M$, Loss Weight $\lambda_{\text{infer}}$, Latent Noise $\epsilon \sim \mathcal{N}(\mathbf{0}, \mathbf{I})$.
2: **for** $i = 1, \ldots, N_o$ **do**
3:      $x_{\text{mix}} \leftarrow x' \odot M + d_{\theta'}(\epsilon) \odot (1 - M)$
4:      $\epsilon \leftarrow \epsilon - \eta_{\text{infer}}\nabla_\epsilon[\|(d_{\theta'}(\epsilon) - x') \odot M\|^2 + \lambda_{\text{infer}}\|\mathcal{R}(x_{\text{mix}})\|^2]$
5: **end for**
6: $x \leftarrow x' \odot M + d_{\theta'}(\epsilon) \odot (1 - M)$
7: **Output** $x$

---

# 3   METHOD: PHYSICS-INFORMED DISTILLATION OF DIFFUSION MODELS

In the previous Section 2, we have demonstrated the existence of the Jensen's Gap when incorporating physical constraints into diffusion training and sampling, as observed in prior works. To address this issue, we propose a distillation-based framework that theoretically bypasses the Jensen's Gap. In specific, instead of enforcing constraints on the posterior mean during the diffusion process which introduces a trade-off with generative accuracy, we apply physical constraints directly to the final generated samples in a post-hoc distillation stage.

## 3.1   DIFFUSION TRAINING

To decouple physical constraint enforcement from the diffusion process itself, we first conduct *standard* diffusion model training using its original denoising objective, without adding any constraint-based loss. To achieve smoother sampling trajectory which benefits later noise-data distillation (39; 40), we adopt a linear diffusion process and apply the $v$-prediction parameterization (39; 37; 40; 8; 16), which is commonly referred to as a flow model. In specific, the training objective is defined as:

$$\mathcal{L}(\boldsymbol{\theta}) = \mathbb{E}_{t\sim U(0,1), x_0\sim p(x_0), \epsilon\sim cN(\mathbf{0},\mathbf{I})} \|(v_\theta(x_t, t) - (\epsilon - x_0)\|^2, \quad x_t = (1-t)x_0 + t\epsilon, \quad (6)$$

where $p(x_0)$ is the distribution of joint data containing both solution and coefficient fields $x = (u, a)$, $\epsilon$ is sampled from a standard Gaussian distribution, and $v_\theta$ is the neural network as the diffusion model. This formulation allows the model to learn to reverse the diffusion process without entangling it with physical supervision, thereby preserving generative fidelity.

## 3.2   IMPOSING PDE CONSTRAINTS IN DISTILLATION

After training the teacher diffusion model using the standard denoising objective, we proceed to the distillation stage, where we transfer its knowledge to a student model designed for efficient one-step

Table 1: Generative metrics on various PDE problems. The PDE error means the MSE of the evaluated physics residual error. The best results are in **bold** and the second best are underlined.

| Dataset | Metric | PIDDM-1 | PIDDM-ref | ECI | DiffusionPDE | D-Flow | PIDM | Teacher |
|---------|--------|---------|-----------|-----|--------------|--------|------|---------|
| Darcy | MMSE ($\times 10^{-2}$) | 0.112 | **0.037** | 0.153 | 0.419 | 0.129 | 0.515 | 0.108 |
| | SMSE ($\times 10^{-2}$) | 0.082 | **0.002** | 0.103 | 0.163 | 0.085 | 0.368 | 0.069 |
| | PDE Error ($\times 10^{-4}$) | 0.226 | **0.148** | 1.582 | 1.071 | 0.532 | 1.236 | 1.585 |
| | FPD | 0.754 | **0.385** | 0.921 | 1.437 | 0.995 | 1.983 | 0.782 |
| | NFE ($\times 10^{3}$) | **0.001** | 0.080 | 0.500 | 0.100 | 5.000 | 0.100 | 0.100 |
| Poisson | MMSE ($\times 10^{-2}$) | 0.162 | **0.113** | 0.183 | 0.861 | 0.172 | 0.948 | 0.150 |
| | SMSE ($\times 10^{-2}$) | 0.326 | **0.274** | 0.291 | 0.483 | 0.475 | 0.701 | 0.353 |
| | PDE Error ($\times 10^{-9}$) | 0.073 | **0.050** | 2.420 | 1.270 | 0.831 | 1.593 | 2.443 |
| | FPD | 1.281 | **0.659** | 1.532 | 1.835 | 1.677 | 2.358 | 1.342 |
| | NFE ($\times 10^{3}$) | **0.001** | 0.080 | 0.500 | 0.100 | 5.000 | 0.100 | 0.100 |
| Burger | MMSE ($\times 10^{-2}$) | 0.152 | **0.012** | 0.294 | 0.064 | 0.305 | 0.948 | 0.264 |
| | SMSE ($\times 10^{-2}$) | 0.133 | **0.101** | 0.105 | 0.103 | 0.207 | 0.701 | 0.114 |
| | PDE Error ($\times 10^{-3}$) | 0.466 | **0.174** | 1.572 | 1.032 | 0.730 | 1.593 | 1.334 |
| | FPD | 0.129 | **0.054** | 0.387 | 1.133 | 0.695 | 1.437 | 0.118 |
| | NFE ($\times 10^{3}$) | **0.001** | 0.080 | 0.500 | 0.100 | 5.000 | 0.100 | 0.100 |

generation. Crucially, this post-hoc distillation stage is where we impose PDE constraints, thereby avoiding the Jensen's Gap observed in prior works that apply constraints during diffusion training or sampling. This distillation process is guided by two complementary objectives: (1) learning to map a noise sample to the final generated output predicted by the teacher model, and (2) enforcing physical consistency on this output via PDE residual minimization. Concretely, we begin by sampling a noise input $\varepsilon \sim \mathcal{N}(0, \mathbf{I})$ and generate a target sample $x_0$ using the pre-trained teacher model via deterministic integration of the reverse-time ODE:

$$\boldsymbol{x}_{t-\mathrm{d}t} = \boldsymbol{x}_t - \boldsymbol{v}_{\boldsymbol{\theta}}(\boldsymbol{x}_t, t)\,\mathrm{d}t, \tag{7}$$

which proceeds from $t = 1$ to $t = 0$ using a fixed step size $dt$. This yields a paired noise-data dataset $\mathcal{D} = \{\boldsymbol{\epsilon}, \boldsymbol{x}_0\}$ for distillation, as shown in Fig. 1 (b). Then a student model $d_{\boldsymbol{\theta}'}(\boldsymbol{\epsilon})$ is trained to predict $\boldsymbol{x}_0$ in one step, as shown in Fig. 1 (c). Meanwhile, to enforce physical consistency, we evaluate the physics residual error on the output $\boldsymbol{x} = d_{\boldsymbol{\theta}'}(\varepsilon)$, i.e., $\|\mathcal{R}(\boldsymbol{x})\|^2$. The overall training objective is:

$$\mathcal{L}_{\text{total}} = \mathcal{L}_{\text{PDE}} + \lambda \mathcal{L}_{\text{sample}} = \mathbb{E}_{(\boldsymbol{\epsilon}, \boldsymbol{x}_0) \sim \mathcal{D}} \left[ \|d_{\boldsymbol{\theta}'}(\boldsymbol{\epsilon}) - \boldsymbol{x}_0\|^2 \right] + \lambda_{\text{train}} \|\mathcal{R}(\boldsymbol{x})\|^2, \tag{8}$$

where $\lambda_{\text{train}}$ balances generative fidelity and physical constraint satisfaction. Unlike prior work (3; 22), this $\lambda_{\text{train}}$ is relatively easy to tune because we mitigate Jensen's Gap by enforcing the PDE directly on samples $x_0 \sim p(x_0 \mid x_t)$ rather than on the posterior mean $\mathbb{E}[x_0 \mid x_t]$ (see Table 3). Training is repeated until convergence (Algorithm 1). This noise-sample distillation is often difficult to learn due to the high curvature of sampling trajectories, yielding noise–data pairs that are far apart in Euclidean space (40). To reduce curvature and improve learnability, we adopt linear-flow distillation (37; 39), and we further evaluate Distribution Matching Distillation (DMD) (70), Rectified Flow (39) and consistency model (60) to strengthen coupling and distribution alignment. These choices produce consistent gains (Table 3).

### 3.3 DOWNSTREAM TASKS

Our method naturally supports one-step generation of physically-constrained data, jointly producing both coefficient and solution fields. Beyond this intrinsic functionality, it also retains the flexibility of the teacher diffusion model, enabling various downstream tasks such as forward and inverse problem solving, and reconstruction from partial observations. Compared to the teacher model, our method achieves these capabilities with improved computational efficiency and stronger physical alignment.

**Generative Modeling.** We aim to sample physically consistent pairs $\boldsymbol{x}_0 = (\boldsymbol{u}, \boldsymbol{a})$ from a learned distribution that satisfies the governing PDE system. The student model supports this via efficient one-step generation: given $\boldsymbol{\epsilon} \sim \mathcal{N}(\mathbf{0}, \mathbf{I})$, it outputs $\boldsymbol{x}_0 = d_{\boldsymbol{\theta}'}(\boldsymbol{\epsilon})$, approximating a valid solution–coefficient pair. We further provide an optional refinement stage based on constraint-driven

optimization (Algorithm 2), which reduces the physics residual by updating $\epsilon$ with gradient descent. This design is inspired by noise prompting methods (4; 18) that optimize the final sample with respect to the initial noise. However, in contrast to those prior work which backpropagate through an entire sampling trajectory and incur high cost and unstable gradients, our refinement operates in a one-step setting. While optional, it offers additional control that is useful in scientific applications requiring strict physical consistency (34; 19; 48; 54).

**Forward/Inverse Problem and Reconstruction.** PIDDM handles all downstream problems as conditional generation over the joint field $\boldsymbol{x} = (\boldsymbol{u}, \boldsymbol{a})$. Forward inference draws $\boldsymbol{u}$ from known $\boldsymbol{a}$; inverse inference recovers $\boldsymbol{a}$ from observed $\boldsymbol{u}$; reconstruction fills in missing entries of $(\boldsymbol{u}, \boldsymbol{a})$ given a partial observation $\boldsymbol{x}'$. We solve this via optimization-based inference on the latent variable $\varepsilon$, using the same student model $d_{\boldsymbol{\theta}'}$ as in generation, as described in Algorithm. 3.. Let $\boldsymbol{x} = d_{\boldsymbol{\theta}'}(\varepsilon)$ denote the generated sample, and let $M$ be a binary observation mask indicating the known entries in $\boldsymbol{x}'$ with respect to $\boldsymbol{x}$. To ensure hard consistency with observed values (e.g., boundary conditions $\mathcal{B}$), we define a mixed sample by injecting observed entries into the generated output, following ECI-sampling (8) and then update $\varepsilon$ by descending the gradient of a combined objective:

$$\mathcal{L}_{\text{total}} = \|(\boldsymbol{x} - \boldsymbol{x}') \odot M\|^2 + \lambda \|\mathcal{R}(\boldsymbol{x}_{\text{mix}})\|^2, \quad \boldsymbol{x}_{\text{mix}} = \boldsymbol{x}' \odot M + \boldsymbol{x} \odot (1 - M). \quad (9)$$

Interestingly, we also find that applying this masking not only enhances hard constraints on $\mathcal{B}$, but also improves satisfaction of $\mathcal{F}$, as demonstrated in our ablation study in Table 3. Classical inverse solvers (35; 36; 43; 51) learn a deterministic map $\boldsymbol{u} \mapsto \boldsymbol{a}$ and therefore require full observations of $\boldsymbol{a}$ to evaluate $\mathcal{F}[\boldsymbol{u}, \boldsymbol{a}] = 0$, a condition rarely met in practice. DiffusionPDE (22) relaxes this by sampling missing variables, but enforces physics on the posterior mean, i.e. $\mathcal{F}\big[\mathbb{E}[\boldsymbol{x}_0 \,|\, \boldsymbol{x}_t]\big]$, and thus suffers from the Jensen's Gap. Our method avoids this inconsistency by imposing constraints directly on the final sample $\mathcal{F}[\boldsymbol{x}_0]$, yielding more reliable and physically consistent inverse solutions.

## 4 EXPERIMENTS

**Experiment Setup.** We consider three widely used PDE benchmarks in main text: Darcy flow, Poisson equation, and Burger's equation. All of these data are readily accessible from FNO (35) and DiffusionPDE (22). We also provide results on other benchmarks in Appendix. E. We consider ECI (8), DiuffsionPDE (22), D-Flow (8; 4), PIDM (3) and vanilla teacher diffusion models as baseline methods, where we put the detailed implementation in Appendix. D.4. We follow ECI-sampling (8) to use FNO as both of the teacher diffusion models and the student distillation model. We put full specification of our experiment setup in Appendix. D.

To quantitatively evaluate generative performance, we report MMSE, SMSE, FPD and PDE error following prior work (8; 30; 3; 27): MMSE measures the mean squared error of the sample mean; SMSE evaluates the error of the sample standard deviation; FPD evaluates Frechet distance between the hidden representations extracted by the pre-trained PDE foundation model; PDE Error quantifies the violation of physical constraints using the physics residual error $|\mathcal{R}(\boldsymbol{x})|^2$. The number of function evaluations (NFE) reflects computational cost during inference. For downstream tasks, we further report MSE on solution, or coefficient fields, or both of them, depending on the problem setting, reflecting the accuracy of PDE solving.

### 4.1 EMPIRICAL EVALUATIONS

PIDDM samples the joint field $(\boldsymbol{u}, \boldsymbol{a})$, enabling forward $(\boldsymbol{u}|\boldsymbol{a})$, inverse $(\boldsymbol{a}|\boldsymbol{u})$, and reconstruction (partial $\boldsymbol{u}, \boldsymbol{a}$) tasks (Sec. 3.3). DiffusionPDE (22) reports only reconstruction MSE, while ECI-sampling (8) and PIDM (3) cover at most one task, limited to either unconditional generation or forward solving. For a fair comparison, we evaluate all methods on all three tasks, providing a unified view of generative quality and physical fidelity.

**Generative Tasks.** We first evaluate the generative performance of our method across three representative PDE systems: Darcy, Poisson, and Burgers' equations. As shown in Table 1, our one-step model (*PIDDM-1*) achieves competitive MMSE and SMSE scores while maintaining extremely low computational cost (1 NFE). Notably, *PIDDM-1 already surpasses all prior methods that incorporate physical constraints during training or sampling*, such as PIDM and DiffusionPDE, ECI-sampling, which suffer from the Jensen's Gap and only exhibit marginal improvements over vanilla diffusion

Table 2: Evaluation on various downstream tasks on Darcy Datasets. The PDE error means the MSE of the evaluated physics residual error. The best results are in **bold**.

| Task | Metric | **PIDDM** | ECI | DiffusionPDE | D-Flow | PIDM |
|------|--------|-----------|-----|--------------|--------|------|
| Forward | MSE ($\times 10^{-1}$) | **0.316** | 0.776 | 0.691 | 0.539 | 0.380 |
| | PDE Error ($\times 10^{-4}$) | **0.145** | 1.573 | 1.576 | 0.584 | 1.248 |
| | NFE ($\times 10^3$) | **0.080** | 0.500 | 0.100 | 5.000 | 0.100 |
| Inverse | MSE ($\times 10^{-1}$) | **0.236** | 0.545 | 0.456 | 0.428 | 0.468 |
| | PDE Error ($\times 10^{-4}$) | **0.126** | 1.505 | 1.402 | 0.438 | 1.113 |
| | NFE ($\times 10^3$) | **0.080** | 0.500 | 0.100 | 5.000 | 0.100 |
| Reconstruct | Coef MSE ($\times 10^{-1}$) | **0.128** | 0.395 | 0.240 | 0.158 | 0.179 |
| | Sol MSE ($\times 10^{-1}$) | **0.102** | 0.219 | 0.143 | 0.125 | 0.147 |
| | PDE Error ($\times 10^{-4}$) | **0.143** | 1.205 | 1.239 | 0.605 | 1.240 |
| | NFE ($\times 10^3$) | **0.080** | 0.500 | 0.100 | 5.000 | 0.100 |

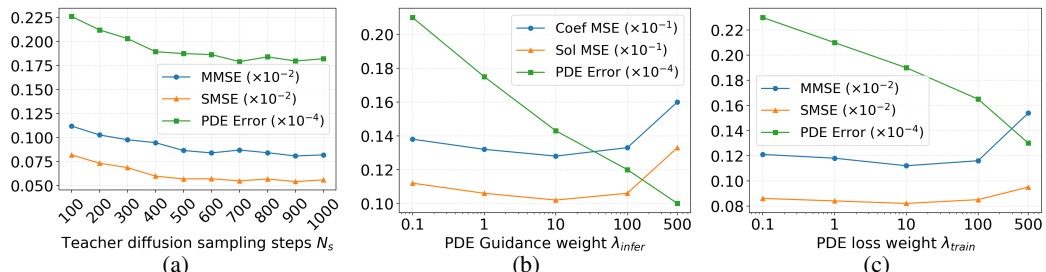

Figure 3: Ablation studies on the effect of several factors on the performance of PIDDM on Darcy dataset. (a), (b) and (c) refer to the effect of the $N_s$, $\lambda_{\text{train}}$ and $\lambda_{\text{infer}}$, respectively.

baselines. *Our optional refinement stage (PIDDM-ref) further reduces both statistical errors and physical PDE residuals, outperforming all baselines.* Meanwhile, ECI—which only enforces hard constraints on boundary conditions—achieves moderate improvements but remains less effective on field-level physical consistency. Although D-Flow theoretically enforces physical constraints throughout the trajectory, it requires thousands of NFEs and often suffers from gradient instability.

**Forward/Inverse Solving and Reconstruction.** We further demonstrate the versatility of our method in forward and inverse problem solving on the Darcy dataset. Since the original PIDM (3) implementation addresses only unconditional generation, we pair it with Diffusion Posterior Sampling (DPS) (9) to extend it to downstream tasks (forward, inverse, and reconstruction). Following the test protocol of D-Flow, we apply inference-time optimization over the initial noise to match given observations while satisfying physical laws. As shown in Table 2, *our method (**PIDDM**) achieves the best results across all metrics*, including MSE and PDE error, while being significantly more efficient than D-Flow, which requires 5000 function evaluations. Compared to ECI and DiffusionPDE, our method yields lower residuals and better predictive accuracy, reflecting its superior handling of physical and observational constraints jointly.

## 4.2 ABLATION STUDIES

To better understand the effect of key design choices in PIDDM, we perform ablations on five factors: teacher sampling steps $N_s$; distillation weight $\lambda_{\text{train}}$; inference weight $\lambda_{\text{infer}}$; diffusion schedule (VP, sub-VP, linear); and advanced distillation variants (Rectified Flow, DMD, Consistency Model). Figure 3 presents three key ablation studies on the Darcy dataset. Panel (a) shows that increasing the teacher model's sampling steps $N_s$ consistently improves both generative quality and physical alignment, as reflected by lower MMSE, SMSE, and PDE residuals of the distilled student, highlighting the importance of high-fidelity supervision. Panels (b) and (c) examine the impact of the PDE loss weight during distillation and inference, respectively. Panels (b) and (c) analyze the effect of the PDE loss weight during inference and distillation, respectively. We observe that across a

Table 3: Evaluation on various downstream tasks on Darcy Datasets on different PIDDM settings. PIDDM: raw method, +RF-1: reflowing (40) for once, +RF-2: reflowing (40) for twice, +DMD: (70), and+CM: (60). −HC refers to the ablation study when not using hard-constraints 3.3 in PIDDM inference. VP and sub-VP refers the ablation study on vp and sub-vp diffusion process. The PDE error means the MSE of the evaluated physics residual error. The best results are in **bold**.

| Task | Metric | PIDDM | +RF-1 | +RF-2 | +DMD | +CM | −HC | VP | sub-VP |
|------|--------|-------|-------|-------|------|-----|-----|-----|--------|
| Forward | MSE ($\times 10^{-1}$) | 0.316 | 0.278 | **0.127** | 0.255 | 0.283 | 0.705 | 0.398 | 0.372 |
| | PDE Error ($\times 10^{-4}$) | 0.145 | 0.129 | 0.098 | 0.134 | **0.083** | 0.354 | 0.154 | 0.157 |
| Inverse | MSE ($\times 10^{-1}$) | 0.236 | 0.195 | **0.136** | 0.188 | 0.182 | 0.503 | 0.284 | 0.271 |
| | PDE Error ($\times 10^{-4}$) | 0.115 | 0.126 | **0.079** | 0.121 | 0.109 | 0.321 | 0.143 | 0.139 |
| Reconstruct | Coef MSE ($\times 10^{-1}$) | 0.128 | 0.107 | 0.091 | 0.095 | **0.085** | 0.294 | 0.133 | 0.138 |
| | Sol MSE ($\times 10^{-1}$) | 0.102 | 0.084 | **0.063** | 0.073 | 0.072 | 0.239 | 0.127 | 0.119 |
| | PDE Error ($\times 10^{-4}$) | 0.143 | 0.118 | 0.085 | 0.104 | **0.83** | 0.464 | 0.159 | 0.158 |

wide range of weights, all metrics achieve strong performance. In particular, compared with baselines in Table 2, the PDE error is reduced by an order of magnitude. This improvement arises because enforcing PDE constraints on the true posterior $x_0 \sim p(x_0 \mid x_t)$ is fundamentally more accurate than on the posterior mean $\mathbb{E}[x_0 \mid x_t]$, thereby theoretically bypassing Jensen's Gap.

We also explore whether more sophisticated distillation strategies can improve the quality of the student model. As shown in Table 3, advanced techniques such as Rectified Flow (RF-1, RF-2), Distribution Matching Distillation (DMD), and Consistency Model yield better MMSE and SMSE than the our raw method, while maintaining competitive PDE residuals. This indicates that tighter coupling between noise and data trajectories during distillation facilitates noise–data learning. Besides, we analyze the effect of imposing hard constraints during downstream inference. Following the strategy inspired by ECI-sampling, we directly replace the masked entries in the generated sample with observed values before computing the PDE residual. This ensures that the known information is preserved when evaluating physical consistency. As shown in Table 3 −HC, cancelling this hard constraint replacement significantly degrades PDE residual across all tasks. We also validate our design on using linear diffusion process in Table 3 VP and sub-VP.

## 5 CONCLUSION AND LIMITATION

**Method.** We introduce **PIDDM**, a lightweight yet effective *post-hoc* distillation framework that enables diffusion models for physics-constrained generation. Concretely, we first train a standard diffusion model and then distill it into a student model by *directly* enforcing PDE constraints on the *final* output. In contrast to existing methods that impose constraints on the posterior mean $\mathbb{E}[x_0 \mid x_t]$, leading to a mismatch known as the *Jensen's Gap* which leads to a trade-off between generative quality and constraint satisfaction, PIDDM applies constraints on the actual sample $x_0$, ensuring physics consistency without sacrificing distributional fidelity.

**Empirical Findings.** We provide the first empirical illustrations of the Jensen's Gap in both diffusion training and sampling, demonstrating its impact on constraint satisfaction. Our experiments show that PIDDM improves both of the physical and distributional fidelity in downstream tasks such as forward, inverse, and partial reconstruction problems with a wide range of hyperparameter choice. Moreover, the student model enables one-step physics simulation, achieving substantial improvements in efficiency while maintaining high physical alignment.

**Limitations.** Our approach assumes access to a well-trained teacher model and a reliable PDE residual operator, which can be difficult to construct, particularly when relying on coarse or low-accuracy finite difference schemes. Moreover, while the one-step student model enables fast inference, its performance may deteriorate if the teacher is poorly calibrated or lacks sufficient trajectory diversity. Although a wide range of PDE loss weights in distillation yields state-of-the-art performance, achieving optimal results still requires tuning. Addressing these limitations is an important direction for future work.

## 6 REPRODUCIBILITY STATEMENT

We have taken several steps to ensure the reproducibility of our results. The full source code, including training and evaluation scripts, is provided in the anonymous supplementary material. Details of the datasets are described in Appendix C, while the training configurations, architectures, and hyperparameters are documented in Appendix D. In addition, we include pseudocode for our proposed algorithms in Alg. 1,2 , and 3 to further clarify the implementation.

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

## A    RELATED WORK

### A.1    DIFFUSION MODEL

Diffusion models (61; 20; 28) learn a score function, $\nabla \log p(\boldsymbol{x}_t)$, to reverse a predefined diffusion process, typically of the form $\boldsymbol{x}_t = \boldsymbol{x}_0 + \sigma_t \boldsymbol{\varepsilon}$. A key characteristic of diffusion models is that sampling requires iteratively reversing this process over a sequence of timesteps. This iterative nature presents a challenge for controlled generation: to guide the sampling trajectory effectively, we often need to first estimate the current denoised target $x_0$ in order to determine the correct guidance direction. In other words, *to decide how to get there, we must first understand where we are*. However, obtaining this information through full iterative sampling is computationally expensive and often impractical in optimization regime.

A practical workaround is to leverage an implicit one-step data estimate provided by diffusion models via the Tweedie's formula (15), which requires only a single network forward pass:
$$\hat{\boldsymbol{x}}_0 \approx \mathbb{E}[\boldsymbol{x}_0 | \boldsymbol{x}_t] = \boldsymbol{x}_t + \sigma_t^2 \nabla \log p(\boldsymbol{x}_t),$$
where $\hat{\boldsymbol{x}}_0$ denotes the final denoised sample from $\boldsymbol{x}_t$ using deterministic sampler. This gap is bridged when $t \to 0$. Although this posterior mean $\mathbb{E}[\boldsymbol{x}_0 \mid \boldsymbol{x}_t]$ is not theoretically equivalent to the final sample obtained after full denoising, in practice, this estimate serves as a useful proxy for the underlying data and enables approximate guidance for controlled generation, without the need to complete the entire sampling trajectory.

### A.2    CONSTRAINED GENERATION FOR PDE SYSTEMS

Diffusion models have demonstrated strong potential for physical-constraint applications due to their generative nature. This generative capability naturally supports the trivial task of simulating physical data and also extends to downstream applications such as reconstruction from partial observations and solving both forward and inverse problems. However, many scientific tasks require strict adherence to physical laws, often expressed as PDE constraints on the data. These constraints, applied at the sample level $\boldsymbol{x}$, are not easily enforced within diffusion models, which are trained to model the data distribution $p(\boldsymbol{x})$. To address this, prior works have proposed three main strategies for incorporating physical constraints into diffusion models.

**Training-time Loss Injection.** PG-Diffusion (56) employs Classifier-Free Guidance (CFG), where a conditional diffusion model is trained using the PDE residual error as a conditioning input. However, CFG is well known to suffer from theoretical inconsistencies—specifically, the interpolated conditional score function does not match the true conditional score—which limits its suitability for enforcing precise physical constraints. To avoid this issue, PIDM (3) introduces a loss term based on the residual evaluated at the posterior mean, $\mathbb{E}[\boldsymbol{x}_0 \mid \boldsymbol{x}_t]$. While this approach avoids the theoretical pitfalls of CFG, the constraint is still not imposed on the actual sample $\boldsymbol{x}_0$, leading to what PIDM identifies as the *Jensen's Gap*.

**Sampling-time Guidance.** Diffusion Posterior Sampling (DPS), used in DiffusionPDE (22) and CoCoGen (27), applies guidance during each sampling step by using the gradient of the PDE residual evaluated on the posterior mean $\mathbb{E}[x_0 \mid x_t]$. Therefore, they inherit the Jensen's Gap issue, as the guidance operates on an estimate of the final sample rather than the sample itself. Moreover, DPS assumes that the residual error follows a Gaussian distribution—a condition that may not hold in real-world PDE systems. Meanwhile, to support hard constraints, ECI-sampling (8) directly modifies the posterior mean $\mathbb{E}[\boldsymbol{x}_0 \mid \boldsymbol{x}_t]$ using known boundary conditions.

**Noise Prompt.** Another stream of research—often called *noise prompting* or *golden-noise optimisation*—directly tunes the *initial* noise so that the resulting sample satisfies a target constraint (4; 18; 75; 67; 45; 7). In the physics domain, this idea is used to minimise the true PDE residual $R(\boldsymbol{x})$ evaluated on the *final* sample, rather than the surrogate residual $R(\mathbb{E}[\boldsymbol{x}_0 \mid \boldsymbol{x}_t])$. Because the constraint is imposed on the actual output, noise prompting sidesteps the Jensen's Gap altogether and therefore serves as a strong baseline in ECI-sampling (8) and PIDM (3). The main drawback is efficiency: optimising the noise requires back-propagating through the entire sampling trajectory, which is computationally expensive and prone to gradient instability.

Recently, diffusion-based techniques for solving image inverse problems have demonstrated competitive performance (72). However, a common limitation is that they rely on the posterior mean, i.e.,

$\mathbb{E}[\boldsymbol{x}_0 \mid \boldsymbol{x}_t]$, as a surrogate for the true posterior. For example, DDRM (29) and DDNM (68) exploit singular value decomposition (SVD) and pseudo-inverse operations to fill in the missing components of $\mathbb{E}[\boldsymbol{x}_0 \mid \boldsymbol{x}_t]$ during sampling, which is conceptually similar to ECI-sampling. Likewise, methods such as DiffPIR (76) and DAPS (71) optimize or run Langevin MCMC updates on the posterior mean in order to enforce observation consistency. Another line of work approximates the likelihood $p(\boldsymbol{y} \mid \boldsymbol{x}_t)$. For instance, DPS (9) treats $p(\boldsymbol{x}_0 \mid \boldsymbol{x}_t)$ as a point mass centered at $\mathbb{E}[\boldsymbol{x}_0 \mid \boldsymbol{x}_t]$, while LGD (59) and DPG (62) use a Gaussian distribution with mean $\mathbb{E}[\boldsymbol{x}_0 \mid \boldsymbol{x}_t]$ for approximation. To reduce this approximation error, some methods trade off computational cost: Monte Carlo–based approaches (14; 6; 23) and variational inference–based approaches (69) avoid the direct mean approximation, but they either require simulating a large number of samples or solving intermediate optimization problems during sampling, both of which are computationally expensive. Finally, there are methods that explicitly aim to reduce the Jensen's Gap by modifying the sampling dynamics. Examples include mid-point schemes (47) and user-defined intermediate potentials (13). While these approaches can shorten the gap, they still rely on $\mathbb{E}[\boldsymbol{x}_0 \mid \boldsymbol{x}_t]$ for posterior estimation, and moreover, they often incur high computational cost due to additional variational inference or Langevin MCMC steps.

### A.3 DISTILLATION OF DIFFUSION MODEL

Sampling in diffusion models involves integrating through a reverse diffusion process, which is computationally expensive. Even with the aid of high-order ODE solvers (46; 58; 41; 42; 74), parallel sampling (55) and better training schedule (28; 49; 38; 39; 40), the process remains iterative and typically requires hundreds of network forward passes. To alleviate this inefficiency, distillation-based methods have been developed to enable one-step generation by leveraging the deterministic nature of samplers (e.g., DDIM), where the noise–data pairs become fixed. The most basic formulation, Knowledge Distillation (44), trains a student model to replicate the teacher's deterministic noise-to-data mapping. However, subsequent studies have shown that directly learning this raw mapping is challenging for neural networks, as the high curvature of sampling trajectories often yields noise–data pairs that are distant in Euclidean space, making the regression task ill-conditioned and hard to generalize.

To address this, recent research has proposed three complementary strategies. (1) Noise–data coupling refinement: Rectified Flow (39) distills the sampling process into a structure approximating optimal transport, where the learned mapping corresponds to minimal-cost trajectories between noise and data. InstaFlow (40) further demonstrates that such near-optimal-transport couplings significantly ease the learning process for student models. (2) Distribution-level distillation: Rather than matching individual noise–data pairs, DMD (70) trains the student via score-matching losses that align the overall data distributions, thereby bypassing the need to regress complex mappings directly. (3) Trajectory distillation: Instead of only supervising on initial ($\boldsymbol{x}_T$) and final ($\boldsymbol{x}_0$) states, this approach provides supervision at intermediate states $x_t$ along the ODE trajectory (5; 73; 60; 63). This decomposition allows the student model to learn the generative process in a piecewise manner, which improves stability and sample fidelity. We note that among existing approaches, Physics-Informed Distillation (PID) (63) bears a similar name to our method but differs fundamentally in both objective and methodology. Specifically, PID distills ODE trajectories from teacher models using a PINN-like strategy, whereas our method distills diffusion models for PDE-constrained generation by applying physical supervision directly to the final samples.

## B  MIXTURE-OF-GAUSSIANS (MOG) DATASET

To study the sampling-time behavior of constrained diffusion models, we design a synthetic 2D Mixture-of-Gaussians (MoG) dataset with analytical score functions. Each sample $x = (x_1, x_2) \in \mathbb{R}^2$ consists of a data dimension $x_1$ and a fixed latent code $x_2$ that serves as a hard constraint.

Specifically, we define a mixture model where $x_1$ is sampled from a Gaussian mixture conditioned on the latent code $z \in \{-1, +1\}$, and $x_2$ is deterministically set to $z$. The full distribution is:

$$x_2 = z \in \{-1, +1\}, \quad x_1 \sim \mathcal{N}(\mu_z, \sigma^2), \tag{10}$$

with $\mu_{-1} = -1$, $\mu_{+1} = +1$, and fixed variance $\sigma^2 = 0.1$. The full 2D data point is thus given by:

$$x = \begin{bmatrix} x_1 \\ x_2 \end{bmatrix}, \quad \text{with } x_1 \sim \mathcal{N}(\mu_{x_2}, \sigma^2), \quad x_2 \in \{-1, +1\}. \tag{11}$$

The resulting joint density $p(x)$ is a mixture of two Gaussians supported on parallel horizontal lines:

$$p(x) = \frac{1}{2}\mathcal{N}(x_1; -1, \sigma^2) \cdot \delta(x_2 + 1) + \frac{1}{2}\mathcal{N}(x_1; +1, \sigma^2) \cdot \delta(x_2 - 1), \tag{12}$$

where $\delta(\cdot)$ denotes the Dirac delta function. In our experiment comparing DPS in Sec. 2.3, we tune the weight of DPS guidance to be 0.035, since it gives satisfying performance.

## B.1 DERIVATION OF SCORE FUNCTION OF THE MoG DATASET

Note that for any MoG, they provide analytical solution of diffusion objectives. In specific, if we consider a MoG with the form:

$$\mathbf{x}_0 \sim \frac{1}{K}\sum_{k=1}^{K}\mathcal{N}(\boldsymbol{\mu}_k, \sigma_k^2 \cdot \boldsymbol{I}),$$

where $K$ is the number of Gaussian components, $\boldsymbol{\mu}_k$ and $\sigma_k^2$ are the means and variances of the Gaussian components, respectively. Suppose the solution of the diffusin process follows:

$$\mathbf{x}_t = \alpha_t \boldsymbol{x}_0 + \sigma_t \cdot \xi \quad \text{where} \quad \xi \sim \mathcal{N}(0, \boldsymbol{I}).$$

Since $\boldsymbol{x}_0$ and $\xi$ are both sampled from Gaussian distributions, their linear combination $\boldsymbol{x}_t$ also forms a Gaussian distribution, i.e.,

$$\mathbf{x}_t \sim \frac{1}{K}\sum_{k=1}^{K}\mathcal{N}(\alpha_t\boldsymbol{\mu}_k, (\sigma_k^2\alpha_t^2 + \sigma_t^2) \cdot \boldsymbol{I}).$$

Then, we have

$$\nabla p_t(\boldsymbol{x}_t) = \frac{1}{K}\sum_{i=1}^{K}\nabla_{\boldsymbol{x}_t}\left[\frac{1}{2}(\frac{1}{\sqrt{2\pi}\sigma_i^2\alpha_t^2 + \sigma_t^2}) \cdot \exp(-\frac{1}{2}(\frac{\boldsymbol{x}_t - \boldsymbol{\mu}_i\alpha_t}{\sigma_i^2\alpha_t^2 + \sigma_t^2})^2)\right]$$

$$= \frac{1}{K}\sum_{i=1}^{K}p_i(\boldsymbol{x}_t) \cdot \nabla_{\boldsymbol{x}_t}\left[-\frac{1}{2}(\frac{\boldsymbol{x}_t - \boldsymbol{\mu}_i\alpha_t}{\sigma_k^2\alpha_t^2 + \sigma_t^2})^2\right]$$

$$= \frac{1}{K}\sum_{i=1}^{K}p_i(\boldsymbol{x}_t) \cdot \frac{-(\boldsymbol{x}_t - \boldsymbol{\mu}_i\alpha_t)}{\sigma_k^2\alpha_t^2 + \sigma_t^2}.$$

We can also calculate the score of $\boldsymbol{x}_t$, i.e.,

$$\nabla \log p_t(\boldsymbol{x}_t) = \frac{\nabla p_t(\boldsymbol{x}_t)}{p_t(\boldsymbol{x}_t)} = \frac{1/K \cdot \sum_{i=1}^{K}p_i(\boldsymbol{x}_t) \cdot \left(\frac{-(\boldsymbol{x}_t - \boldsymbol{\mu}_i\alpha_t)}{\sigma_k^2\alpha_t^2 + \sigma_t^2}\right)}{1/K \cdot \sum_{i=1}^{K}p_i(\boldsymbol{x}_t)}.$$

## B.2 DEVIATION OF VELOCITY FIELD OF REVERSE ODE AND DPS

Diffusion models define a forward diffusion process to perturb the data distribution $p_{data}$ to a Gaussian distribution. Formally, the diffusion process is an Itô SDE $\mathrm{d}\boldsymbol{x}_t = \boldsymbol{f}(\boldsymbol{x}_t) + g(t)\mathrm{d}\mathbf{w}$, where $\mathrm{d}\mathbf{w}$ is the Brownian motion and $t$ flows forward from $0$ to $T$. The solution of this diffusion process gives a transition distribution $p_t(\boldsymbol{x}_t|\boldsymbol{x}_0) = \mathcal{N}(\boldsymbol{x}_t|\alpha_t\boldsymbol{x}_0, \sigma_t^2\mathbf{I})$, where $\alpha_t = e^{\int_0^t f(s)ds}$ and $\sigma_t^2 = 1 - e^{-\int_0^t g(s)^2 ds}$. Specifically in linear diffusion process, $\alpha_t = t$, and $\beta_t = 1 - t$. To sample from the diffusion model, a typical approach is to apply a reverse-time SDE which reverses the diffusion process (2):

$$\mathrm{d}\boldsymbol{x}_t = [\boldsymbol{f}(\boldsymbol{x}_t) - g(t)^2\nabla_{\boldsymbol{x}_t}\log p_t(\boldsymbol{x}_t)]\mathrm{d}t + \mathrm{d}\bar{\mathbf{w}},$$

where $\mathrm{d}\bar{\mathbf{w}}$ is the Brownian motion and $t$ flows forward from $T$ to $0$. For all reverse-time SDE, there exists corresponding deterministic processes which share the same density evolution, i.e., $\{p_t(x_t)\}_{t=0}^T$ (61). In specific, this deterministic process follows an ODE:

$$\mathrm{d}\boldsymbol{x}_t = [\boldsymbol{f}(\boldsymbol{x}_t) - \frac{1}{2}g(t)^2\nabla_{\boldsymbol{x}_t}\log p_t(\boldsymbol{x}_t)]\mathrm{d}t,$$

where $t$ flows backwards from $T$ to $0$. The deterministic process defines a velocity field,

$$v_{\text{GT}}(\boldsymbol{x}, t) = [\boldsymbol{f}(\boldsymbol{x}_t) - \frac{1}{2}g(t)^2 \nabla_{\boldsymbol{x}_t} \log p_t(\boldsymbol{x}_t)].$$

Here, we also define the velocity field by $v(\boldsymbol{x}_t, t) = \boldsymbol{f}(\boldsymbol{x}_t) - \frac{1}{2}g(t)^2 \nabla_{\boldsymbol{x}_t} \log p_t(\boldsymbol{x}_t)$.

The posterior mean can be estimated from score by:

$$\mathbb{E}[\boldsymbol{x}_0 | \boldsymbol{x}_t]) = \frac{\boldsymbol{x}_t + \sigma_t^2 \nabla \log p_t(\boldsymbol{x}_t)}{\alpha_t}.$$

The posterior mean can be also estimated from velocity field by:

$$\mathbb{E}[\boldsymbol{x}_0 | \boldsymbol{x}_t]) = \frac{\dot{\sigma} \boldsymbol{x}_t - \sigma_t v(\boldsymbol{x}_t, t)}{\alpha_t \dot{\sigma}_t + \sigma_t \dot{\alpha}_t}.$$

## C   DATASETS

### C.1   DARCY FLOW

We adopt the Darcy Flow setup introduced in DiffusionPDE (22) and the dataset is released from FNO (35). For completeness, we describe the generation process here. In specific, we consider the steady-state Darcy flow equation on a 2D rectangular domain $\Omega \subset \mathbb{R}^2$ with no-slip boundary conditions:

$$-\nabla \cdot (a(c)\nabla u(c)) = q(c), \quad c \in \Omega, \quad u(c) = 0, \quad c \in \partial\Omega.$$

Here, $a(c)$ is the spatially varying permeability field with binary values, and $q(c)$ is set to 1 for constant forcing. The $(u, a)$ is jointly modeled by diffusion model.

### C.2   INHOMOGENEOUS HELMHOLTZ EQUATION AND POISSON EQUATION

We adopt the setup introduced in DiffusionPDE (22) and the dataset is released from FNO (35). For completeness, we describe the generation process here. As a special case of the inhomogeneous Helmholtz equation, the Poisson equation is obtained by setting $k = 0$:

$$\nabla^2 u(c) = a(c), \quad c \in \Omega, \quad = u(c) = 0, \quad c \in \partial\Omega.$$

Here, $a(c)$ is a piecewise constant forcing function. The $(u, a)$ is jointly modeled by diffusion model.

### C.3   BURGERS' EQUATION

We adopt the Burgers' Equation setup introduced in DiffusionPDE (22) and the dataset is released from FNO (35). For completeness, we describe the generation process here. We study the 1D viscous Burgers' equation with periodic boundary conditions on a spatial domain $\Omega = (0, 1)$ and temporal domain $\tau \in (0, T]$:

$$\partial_\tau u(c, \tau) + \partial_c \left( \frac{u^2(c, \tau)}{2} \right) = \nu \partial_{cc}^2 u(c, \tau), \quad u(c, 0) = a(c), \quad c \in \Omega.$$

In our experiments, we set $\nu = 0.01$. Specifically, we use 128 temporal steps, where each trajectory has shape $128 \times 128$. The $(u, a)$ is jointly modeled by diffusion model.

### C.4   STOKES PROBLEM

We adopt the Stokes problem setup introduced in ECI-Sampling (8) and use their released generation code. For completeness, we describe the generation process below.

The 1D Stokes problem is governed by the heat equation:

$$u_t = \nu u_{xx}, \quad x \in [0, 1], \ t \in [0, 1],$$

with the following boundary and initial conditions:

$$u(x,0) = Ae^{-kx}\cos(kx), \quad x \in [0,1], \quad a(t) := u(0,t) = A\cos(\omega t), \quad t \in [0,1],$$

where $\nu \geq 0$ is the viscosity, $A > 0$ is the amplitude, $\omega$ is the oscillation frequency, and $k = \sqrt{\omega/(2\nu)}$ controls the spatial decay. The analytical solution is given by:

$$u_{\text{exact}}(x,t) = Ae^{-kx}\cos(kx - \omega t).$$

In our experiments, we set $A = 2$, $k = 5$ and take $a := \omega \sim \mathcal{U}[2,8]$ as the coefficient field to jointly model with $u$.

## C.5    HEAT EQUATION

We adopt the heat equation setup introduced in ECI-Sampling (8) and use their released generation code. For completeness, we describe the generation process below.

The 1D heat (diffusion) equation with periodic boundary conditions is defined as:

$$u_t = \alpha u_{xx}, \quad x \in [0, 2\pi], \ t \in [0,1],$$

with the initial and boundary conditions:

$$a(x) := u(x,0) = \sin(x + \varphi), \quad a(t) := u(0,t) = u(2\pi, t).$$

Here, $\alpha$ denotes the diffusion coefficient and $\varphi$ controls the phase of the sinusoidal initial condition. The exact solution is:

$$u_{\text{exact}}(x,t) = e^{-\alpha t}\sin(x + \varphi).$$

In our experiments, we set $\alpha = 3$ and take $a := \varphi \sim \mathcal{U}[0,\pi]$ as the coefficient to jointly model with $u$.

## C.6    NAVIER–STOKES EQUATION

We adopt the 2D Navier–Stokes (NS) setup from ECI-Sampling (8) and use their released generation code. The NS equation in vorticity form for an incompressible fluid with periodic boundary conditions is given as:

$$\partial_t w(x,t) + u(x,t) \cdot \nabla w(x,t) = \nu \Delta w(x,t) + f(x), \quad x \in [0,1]^2, \ t \in [0,T],$$
$$\nabla \cdot u(x,t) = 0, \quad x \in [0,1]^2, \ t \in [0,T],$$
$$w(x,0) = w_0(x), \quad x \in [0,1]^2.$$

Here, $u$ denotes the velocity field and $w = \nabla \times u$ is the vorticity. The initial vorticity $w_0$ is sampled from $\mathcal{N}(0, 7^{3/2}(-\Delta + 49I)^{-5/2})$, and the forcing term is defined as $f(x) = 0.1\sqrt{2}\sin(2\pi(x_1 + x_2) + \phi)$, where $\phi \sim U[0, \pi/2]$. We take $a := w_0$ as the coefficient to jointly model with $u$.

## C.7    POROUS MEDIUM EQUATION

We use the Porous Medium Equation (PME) setup provided by ECI-Sampling (8), with zero initial and time-varying Dirichlet left boundary conditions:

$$u_t = \nabla \cdot (u^m \nabla u), \quad x \in [0,1], \ t \in [0,1],$$
$$(x,0) = 0, \quad x \in [0,1],$$
$$u(0,t) = (mt)^{1/m}, \quad t \in [0,1],$$
$$u(1,t) = 0, \quad t \in [0,1].$$

The exact solution is $u_{\text{exact}}(x,t) = (m \cdot \text{ReLU}(t - x))^{1/m}$. The exponent $m$ is sampled from $U[1,5]$. We take $a := m$ as the coefficient to jointly model with $u$.

## C.8 STEFAN PROBLEM

We also adopt the Stefan problem configuration from ECI-Sampling (8), which is a nonlinear case of the Generalized Porous Medium Equation (GPME) with fixed Dirichlet boundary conditions:

$$u_t = \nabla \cdot (k(u)\nabla u), \quad x \in [0,1], \, t \in [0,T],$$
$$a(x,0) := u(x,0) = 0, \quad x \in [0,1],$$
$$a(0,t) := u(0,t) = 1, \quad t \in [0,T],$$
$$a(1,t) := u(1,t) = 0, \quad t \in [0,T],$$

where $k(u)$ is a step function defined by a shock value $u^*$:

$$k(u) = \begin{cases} 1, & u \geq u^*, \\ 0, & u < u^*. \end{cases}$$

The exact solution is:

$$u_{\text{exact}}(x,t) = \mathbb{1}_{[u \geq u^*]} \left( 1 - (1-u^*) \frac{\text{erf}(x/(2\sqrt{t}))}{\text{erf}(\alpha)} \right),$$

where $\alpha$ satisfies the nonlinear equation $(1-u^*)/\sqrt{\pi} = u^* \, \text{erf}(\alpha)\alpha \exp(\alpha^2)$. We follow ECI-Sampling to take $a := u^* \sim U[0.55, 0.7]$ as the coefficient to jointly model with $u$.

# D  EXPERIMENTAL SETUP

This section provides details on the model architecture, training configurations for diffusion and distillation, evaluation protocols, and baseline methods.

## D.1  MODEL STRUCTURE

We follow ECI-sampling (8) and adopt the Fourier Neural Operator (FNO) (35) as both the teacher diffusion model and the student distillation model. A sinusoidal positional encoding (65) is appended as an additional input dimension. Specifically, we use a four-layer FNO with a frequency cutoff of $32 \times 32$, a time embedding dimension of 32, a hidden channel width of 64, and a projection dimension of 256.

## D.2  DIFFUSION AND DISTILLATION TRAINING SETUP

For diffusion training, we employ a standard linear noise schedule (39; 38; 37; 40) with a batch size of 128 and a total of 10,000 iterations. The model is optimized using Adam (31) with a learning rate of $3 \times 10^{-2}$.

During distillation, we use Euler's method with 100 uniformly spaced timesteps from $t = 1$ to $t = 0$ for sampling. Every 100 epochs, we resample 1024 new noise–data pairs for supervision. Distillation is trained for 2000 epochs using Adam (learning rate $3 \times 10^{-2}$), with early stopping based on the squared norm of the observation loss, i.e., $\|s_{\theta'}(\varepsilon) - x\|^2$.

The physics constraint weight $\lambda_{\text{train}}$ is set to 10 for Darcy Flow, Burgers' Equation, Stokes Flow, Heat Equation, Navier–Stokes, Porous Medium Equation, and Stefan Problem. For Helmholtz and Poisson equations, we increase $\lambda_{\text{train}}$ to $10^6$ due to the stiffness of these PDEs. All experiments are conducted on an NVIDIA RTX 4090 GPU.

## D.3  EVALUATION SETUP

For physics-based data simulation, we evaluate models with and without physics refinement: the number of gradient-based refinement steps $N$ is set to 0 or 50. The step size $\eta$ is aligned with the dataset-specific $\lambda_{\text{train}}$ used during distillation.

In forward and inverse problems, the observation mask $M$ defines the known entries. For forward problems, the mask has ones at boundary entries. For partial reconstruction, the mask is sampled randomly with 20% of entries set to 1 (observed), and the rest to 0 (missing). All evaluations are conducted on an NVIDIA RTX 4090 GPU.

## D.4 BASELINE METHODS

We describe the configurations of all baseline methods used for comparison. Where necessary, we adapt our diffusion training and sampling codebase to implement their respective constraint mechanisms.

**ECI-sampling.** We follow the approach of directly substituting hard constraints onto the posterior mean $\mathbb{E}[x_0 \mid x_t]$ based on a predefined observation mask. Specifically, we project these constraints at each DDIM step (58) using a correction operator $C$:

$$\boldsymbol{x}_{t-dt} = C(\hat{\boldsymbol{x}}_\theta(\boldsymbol{x}_t, t)) \cdot (1 - t + \mathrm{d}t) + (\boldsymbol{x}_t - \hat{\boldsymbol{x}}_\theta(\boldsymbol{x}_t, t)) \cdot (t - \mathrm{d}t), \tag{13}$$

where $t$ flows backward from 1 to 0, and $\hat{\boldsymbol{x}}_\theta$ denotes the posterior mean estimated using Tweedie's formula.

**DiffusionPDE.** This method employs diffusion posterior sampling (DPS) (9), where each intermediate sample $\boldsymbol{x}_t$ is guided by the gradient of the PDE residual evaluated on the posterior mean:

$$\boldsymbol{x}_{t-dt} = \boldsymbol{x}_t + v_\theta(\boldsymbol{x}_t, t) \cdot \mathrm{d}t - \eta_t \nabla_{\boldsymbol{x}_t} \|\mathcal{R}(\hat{\boldsymbol{x}}_\theta(\boldsymbol{x}_t, t))\|^2, \tag{14}$$

where $v_\theta(\boldsymbol{x}_t, t)$ is the learned velocity field from the reverse-time ODE sampler, and $\eta_t$ is a hyperparameter. In our experiments, we set $\eta_t$ equal to $\lambda_{\text{train}}$ for each dataset.

**PIDM.** This method incorporates an additional residual loss into the diffusion training objective, evaluated on the posterior mean $\mathbb{E}[x_0 \mid x_t]$. Specifically, PIDM (3) augments the standard diffusion loss with a physics-based term:

$$\mathcal{L}_{\text{PIDM}} = \mathcal{L}_{\text{diffusion}} + \lambda_t \|\mathcal{R}(\hat{\boldsymbol{x}}_\theta(\boldsymbol{x}_t, t))\|^2, \tag{15}$$

where $\mathcal{L}_{\text{diffusion}}$ is the original diffusion training loss, and $\lambda_t$ is the residual loss weight. In our experiment, we set $\lambda_t$ to be $\lambda_{\text{train}}$ for each dataset since it gives satisfying performance.

**D-Flow.** For this standard method (4), We build on the official implementation of ECI-sampling (8) and introduce an additional PDE residual loss evaluated on the final sample. The weighting $\lambda_{\text{train}}$ is aligned with our setup across datasets. Specifically, the implementation follows the D-Flow setup in ECI-sampling (8): we discretize the sampling trajectory into 100 denoising steps and perform gradient-based optimization on the input noise over 50 iterations to minimize the physics residual loss. At each iteration, gradients are backpropagated through the entire 100-step trajectory, resulting in a total of 50,000 function evaluations (NFE) per sample. This leads to significantly higher computational cost compared to our one-step method.

**Teacher.** This baseline refers to sampling directly from the trained teacher diffusion model without incorporating any PDE-based constraint or guidance mechanism.

# E  GENERATIVE EVALUATIONS ON MORE DATASETS

In this section, we include the performance of our results on more datasets and comparisons to other baseline methods, as shown in Table. 4. PIDDM marginally surpass all baseline especially in the physics residual error.

Table 4: Generative metrics on various constrained PDEs. The PDE error means the MSE of the evaluated physics residual error. The best results are in **bold**.

| Dataset | Metric | PIDDM-1 | PIDDM-ref | ECI | DiffusionPDE | D-Flow | PIDM | FM |
|---|---|---|---|---|---|---|---|---|
| Helmholtz | MMSE ($\times 10^{-1}$) | 0.265 | **0.185** | 0.318 | 0.335 | 0.140 | 0.352 | 0.296 |
| | SMSE ($\times 10^{-1}$) | 0.195 | **0.169** | 0.289 | 0.301 | 0.106 | 0.325 | 0.210 |
| | PDE Error ($\times 10^{-9}$) | 0.054 | **0.034** | 2.135 | 1.812 | 0.680 | 1.142 | 2.104 |
| | NFE ($\times 10^{3}$) | **0.001** | 0.100 | 0.500 | 0.100 | 5.000 | 0.100 | 0.100 |
| Stokes Problem | MMSE ($\times 10^{-2}$) | 0.298 | **0.182** | 0.335 | 0.342 | 0.301 | 0.361 | 0.310 |
| | SMSE ($\times 10^{-2}$) | 0.425 | **0.312** | 0.455 | 0.469 | 0.441 | 0.484 | 0.430 |
| | PDE Error ($\times 10^{-3}$) | 0.241 | **0.194** | 0.585 | 0.498 | 0.318 | 0.432 | 0.578 |
| | NFE ($\times 10^{3}$) | **0.001** | 0.100 | 0.500 | 0.100 | 5.000 | 0.100 | 0.100 |
| Heat Equation | MMSE ($\times 10^{-3}$) | 0.901 | **0.845** | 4.620 | 4.600 | 1.452 | 4.580 | 4.544 |
| | SMSE ($\times 10^{-2}$) | 0.816 | **0.790** | 1.612 | 1.598 | 0.892 | 1.587 | 1.565 |
| | PDE Error ($\times 10^{-5}$) | 3.265 | **2.910** | 4.120 | 4.100 | 3.698 | 4.150 | 4.354 |
| | NFE ($\times 10^{3}$) | **0.001** | 0.100 | 0.500 | 0.100 | 5.000 | 0.100 | 0.100 |
| Navier–Stokes Equation | MMSE ($\times 10^{-4}$) | 0.285 | **0.264** | 0.302 | 0.299 | 0.288 | 0.306 | 0.294 |
| | SMSE ($\times 10^{-4}$) | 0.218 | **0.210** | 0.323 | 0.321 | 0.225 | 0.327 | 0.314 |
| | PDE Error ($\times 10^{-5}$) | 3.184 | **2.945** | 6.910 | 6.740 | 3.200 | 6.950 | 7.222 |
| | NFE ($\times 10^{3}$) | **0.001** | 0.100 | 0.500 | 0.100 | 5.000 | 0.100 | 0.100 |
| Porous Medium Equation | MMSE ($\times 10^{-3}$) | 4.555 | **4.210** | 7.742 | 7.698 | 5.203 | 7.762 | 7.863 |
| | SMSE ($\times 10^{-1}$) | 2.143 | **2.051** | 2.573 | 2.602 | 2.327 | 2.589 | 2.639 |
| | PDE Error ($\times 10^{-5}$) | 3.412 | **3.110** | 4.982 | 4.945 | 3.548 | 4.917 | 5.523 |
| | NFE ($\times 10^{3}$) | **0.001** | 0.100 | 0.500 | 0.100 | 5.000 | 0.100 | 0.100 |
| Stefan Problem | MMSE ($\times 10^{-3}$) | 0.231 | **0.220** | 0.248 | 0.249 | 0.238 | 0.252 | 0.245 |
| | SMSE ($\times 10^{-3}$) | 0.278 | **0.268** | 0.315 | 0.318 | 0.289 | 0.320 | 0.307 |
| | PDE Error ($\times 10^{-2}$) | 0.081 | **0.070** | 0.410 | 0.398 | 0.095 | 0.405 | 0.458 |
| | NFE ($\times 10^{3}$) | **0.001** | 0.100 | 0.500 | 0.100 | 5.000 | 0.100 | 0.100 |

# F    QUALITATIVE RESULTS ON THE DARCY FORWARD PROBLEM

Figure 4 compares the predicted Darcy pressure fields and their corresponding data- and PDE-error maps for each baseline and for our PIDDM. DiffuionPDE, and ECI reproduce the coarse flow pattern but exhibit large point-wise errors and pronounced residual bands. In contrast, PIDDM produces the visually sharpest solution and the lowest error intensities in both maps, confirming the quantitative gains reported in the main text.

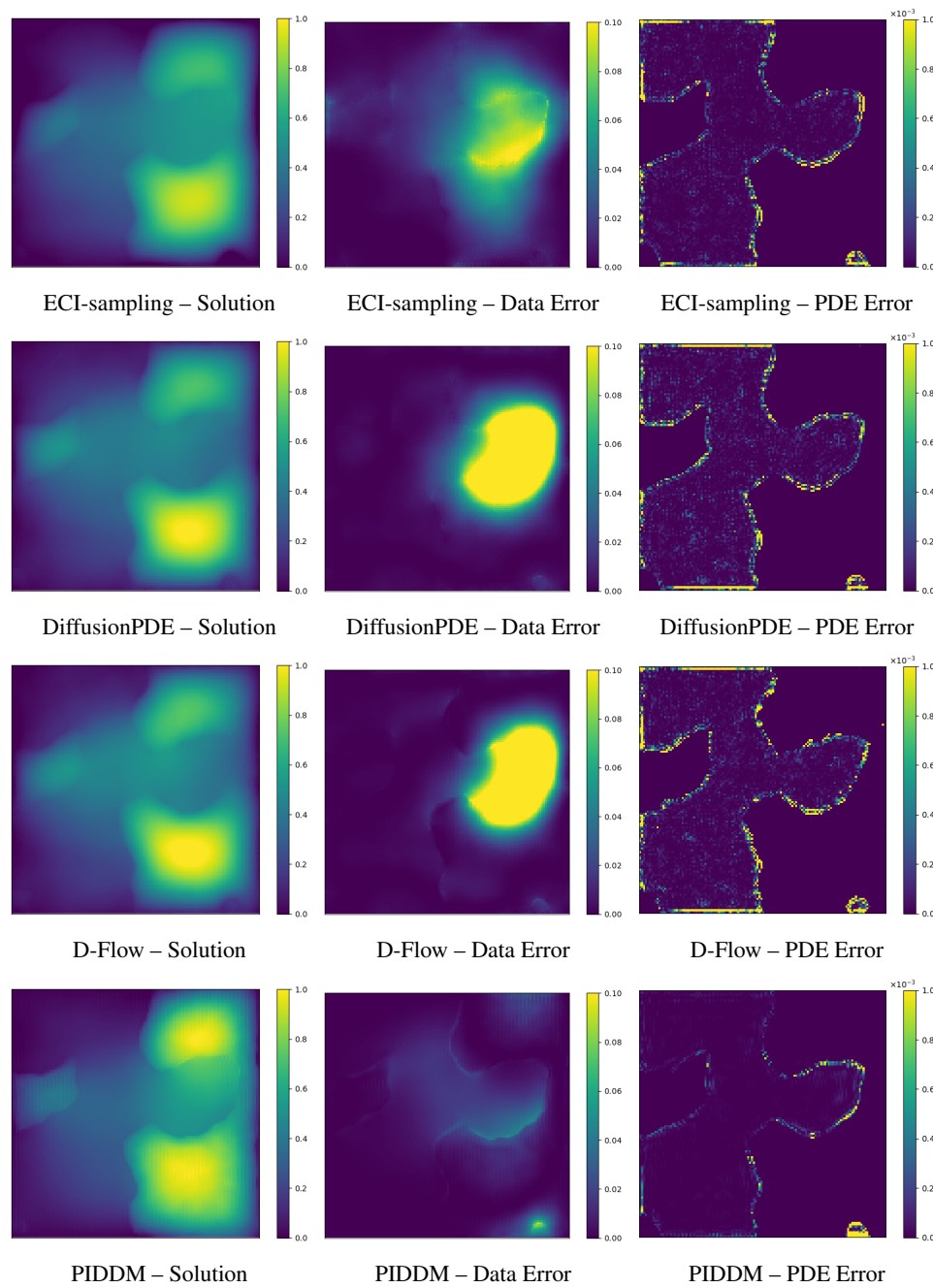

Figure 4: Qualitative comparison on the Darcy *forward* problem. Each column shows (left) the predicted solution field, (middle) point-wise data error, and (right) PDE residual error. Our PIDDM (bottom row) delivers visibly lower data and PDE errors than other baselines while maintaining sharp solution details.

## G    ADDITIONAL EXPERIMENTS

We investigate a controlled Mixture-of-Gaussians (MoG) setting to evaluate constraint satisfaction in generative models. The target distribution is a correlated, two-component Gaussian mixture:

$$p_{\mathrm{MoG}}(\mathbf{x}) = \tfrac{1}{2}, \mathcal{N}\left(\mathbf{x}; [-1, -1]^\top, \Sigma\right) + \tfrac{1}{2}, \mathcal{N}\left(\mathbf{x}; [+1, +1]^\top, \Sigma\right), \tag{16}$$

where $\Sigma = \sigma^2 \begin{bmatrix} 1 & \rho \\ \rho & 1 \end{bmatrix}, \sigma^2 = 0.04, \rho = 0.99999$. The high correlation $\rho = 0.99999$ ensures that the analytic score function $\nabla_{\mathbf{x}} \log p_{\mathrm{MoG}}(\mathbf{x})$ remains well-defined, despite the near-singular covariance. The physical constraint is defined as:

$$\mathcal{F}(\boldsymbol{x}) = |\boldsymbol{x}_0 - \boldsymbol{x}_1|^2 = 0. \tag{17}$$

**Baselines.** DPS and ECI both integrate the analytical score using 1000-step Euler discretization over $(0, 1)$. DPS applies constraint guidance via gradient descent on $\mathcal{F}(\boldsymbol{x})$ at each step, using a loss weight of 300. ECI enforces the constraint by directly projecting the posterior mean to satisfy $\mathcal{F}(\boldsymbol{x}) = 0$.

**PIDDM.** A teacher diffusion model is constructed using a probability-flow ODE with 100-step Euler integration, leveraging the analytic score. It generates 50,000 training pairs $(\boldsymbol{\varepsilon}, \boldsymbol{x}_0)$ which are used to train a one-step student model, a ReLU-activated MLP with two hidden layers (100 neurons each) via the loss:

$$\mathcal{L}_{\mathrm{train}} = \frac{1}{N} \sum_{i=1}^{N} |d_{\boldsymbol{\theta}}(\boldsymbol{\varepsilon}_i) - \boldsymbol{x}_{0,i}|^2 + \lambda_{\mathrm{train}} \mathcal{F}(d_{\boldsymbol{\theta}}(\boldsymbol{\varepsilon}_i)), \quad \lambda_{\mathrm{train}} = 1. \tag{18}$$

Training uses Adam optimizer ($\mathrm{lr} = 10^{-3}$, batch size = 2048). During inference, latent noise $\boldsymbol{\varepsilon}$ is optimized via 80 steps of LBFGS with strong-Wolfe line search, learning rate $3 \times 10^{-3}$, and gradient tolerance $10^{-7}$, with $\lambda_{\mathrm{infer}} = 1$.

**Results.** Figure 5(a) shows that all methods recover the bimodal structure of $\boldsymbol{x}_1$. However, as shown in Figure 5(b), DPS fails to fully satisfy the constraint, with $\boldsymbol{x}_0 - \boldsymbol{x}_1$ spread over $\pm 10^{-2}$, while ECI enforces it exactly but distorts the marginal distribution. In contrast, PIDDM maintains both constraint satisfaction (standard deviation $\approx 2 \times 10^{-3}$) and distributional fidelity.

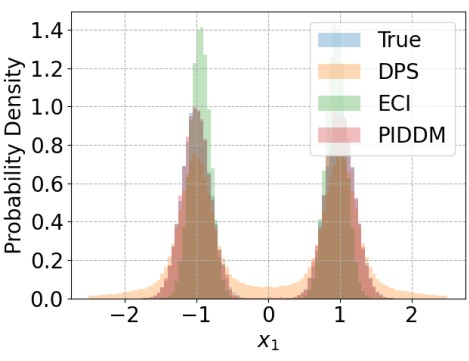
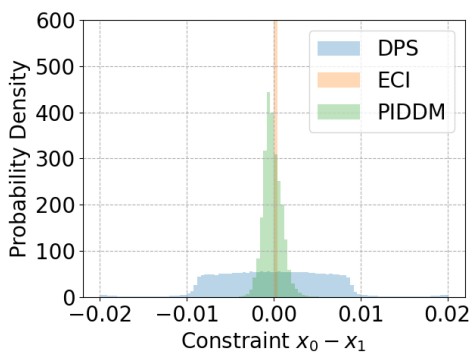

(a) Marginal distribution over $x_1$         (b) Constraint deviation $x_0 - x_1$

Figure 5: Constraint satisfaction on correlated MoG. Comparison of generated samples using DPS, ECI, and PIDDM. PIDDM closely matches the target distribution while satisfying constraints.

## H    USE OF LARGE LANGUAGE MODELS

We used large language models (LLMs) solely as an assistive tool for polishing the writing and improving clarity of exposition. LLMs were not involved in research ideation, experiment design, implementation, or analysis.

