# OpenReview forum: "Physics-Informed Distillation of Diffusion Models for PDE-Constrained Generation"
_ICLR.cc/2026/Conference — ICLR 2026 Conference Withdrawn Submission_

### Official Review · Reviewer_ZQ1z · 2025-10-22

**Soundness:** 2
**Presentation:** 4
**Contribution:** 2
**Rating:** 4
**Confidence:** 4

**Summary:**

The paper tackles a core mismatch in physics-guided diffusion: prior methods enforce PDE constraints on the posterior mean at noisy states, which doesn’t guarantee the final samples obey the physics (a Jensen-gap issue). It proposes Physics-Informed Distillation (PIDDM): train a standard teacher diffusion model, then distill a one-step student and penalize the PDE residual directly on the student’s outputs; at test time, a tiny latent-noise refinement can further reduce residuals. The same student supports forward, inverse, and partial-reconstruction by optimizing its latent under data masks plus the PDE residual. On Darcy/Poisson/Burgers, PIDDM achieves lower PDE error and competitive or better MSE than prior physics-aware baselines, while using ~1 NFE. The key contribution is a simple recipe that enforces physics on final samples to avoid the Jensen gap and cutting compute.

**Strengths:**

- Originality. This paper takes a simple but fresh angle: stop enforcing physics on noisy states on posterior means and put the PDE loss on the actual final samples. Doing this via teacher -> one step student distillation with PDE loss is a clean combo. It is not a brand-new primitive but a well-targeted rethink of where the constraints belongs.

- Quality. The paper supports its claims across some PDE benchmarks, and comprarisons are made against some of the relevant and competitive baselines. The results consistently demonstrate PIDDM's effectiveness. Extensive ablation studies further investigate key design choices.

- Clarity. This paper clearly articulates and empirically shows the issue of Jensen's Gap. The motivation is strong. The objective and algorithms are readable.

- Significance. It enables one-step unconditional generation (NFE=1), and the same distilled student model can be adapted via inference-time optimization to handle different (but inherently the same under the paper's framing) problems.

**Weaknesses:**

### Unjustified Complexity for Downstream Tasks Compared to Standard Inverse Problem Methods
The paper's approach to downstream tasks uses a complex, iterative optimization that appears potentially redundant, relatively fragile, and poorly justified given the problem's inherent structure and existing solution paradigms.
- By modeling the joint filed $x=(u,a)$, all downstream tasks inherently become inverse problems: estimating the unknown components of $x$ given partial observations defined by mask $M$ and governing physics $\mathcal{R}(x)=0$. Whether the known part is $a$ (forward) $x$ (inverse), or a subset of both (reconstuction) is merely a difference in the mask M.
- For the primary downstream tasks, the method (Algorithm 3) optimizes $\varepsilon$ in the latent space that has the same dimensionality as the field, and backpropagates through both a student model distilled from a pre-trained teacher model, and the discrete residual implemented implemented in a differentiable way.
- Since the method already requires a differentiable implementation of the physics residual $\mathcal{R}(x)$, one could directly solve these problems via standard gradient-based optimization in the data space. The objective would simply minimize a combination of the data mismatch term and the physics residual term
$$\min_x ||(x-x^\prime)\odot M||^2 + \lambda ||\mathcal{R}(x\odot(1-M)+x^\prime\odot M)||^2.$$
This avoids the need for any model (no teacher, no student) and optimizes directly on the quantity of interest. The paper does not provide theoretical argument or empirical comparison to demonstrate that Algorithm 3 offers advantages (e.g., better convergence, finding better minima, improved sample quality) over the simpler, direct optimization approach. The significant overhead of training the diffusion and student model seems entirely unjustified without demonstrating superiority over simpler optimization baselines.
- The authors acknowlege that the work  assumes access to a well-trained teacher model, but even assuming a perfectly trained teacher and student, optimizing through the high-dimensional latent space via the complex, learned student inherently complicates the optimization landscape.
- More importantly, direct optimization relies primarily on the physics model and observed data without concerns about out-of-distribution generalization. In contrast, PIDDM relies heavily on the student model. Given the infinite dimensionality of PDE solution spaces and infinite choices of parameters/initial conditions/boundary conditions, it is highly likely that the teacher model (and thus the student) will be trained on insufficient or non-reprensentative data. If the specific problem instance falls outside the distribution learned by the teacher/student, the student model could actively harm the optimization.

- **This questions the utility of the proposed framework for anything beyond unconditional generation.**

### Unclear statistical significance
The paper compares PIDDM against several baselines using metrics like MMSE, SMSE, FPD, and PDE error, reporting single numerical values for each method and dataset (Tables 1, 2, 3, 4). However, as these are generative models, their outputs are inherently stochastic, depending on factors like the initial noise seed, and potentially the optimization path in Algorithm 3. Without error bars, it is impossible to assess the statistical significance of the reported differences between methods. This is particularly problematic where the performance metrics between PIDDM and certain baselines are very close. The authors are encouraged to report uncertainty calculated over multiple independent runs for all reported metrics to strengthen the paper's claims.

**Questions:**

### Questions/Confusions
- **On "bypassing Jensen's gap via the true posterior $p(x_0|x_t)$" claim**. In section 4.2, the paper claims its improvement arises from *"enforcing PDE constraints on the true posterior $x_0 \sim p(x_0|x_t)$, which is fundamentally more accurate than on the posterior mean $\mathbb{E}(x_0|x_t)$, thereby theoretically bypassing Jensen's Gap"*. However, the method described in Algorithm 1 and Equation (8) applies the PDE loss to the output of the student model . This output is a single, deterministic function of ϵ, trained to approximate samples from the teacher. Is it not a significant overstatement to call this "enforcing constraints on the true posterior"? The method appears to constrain a single, learned approximation of a sample, not the true (and intractable) posterior distribution. Could the authors please clarify this discrepancy in the theoretical framing?

### Errors in the tables.
In Table 1: ECI, instead of PIDDM-1, is the second best in terms of SMSE for Poisson equation.   Diffusion PDE, instead of PIDDM-1, is the second best in terms of MMSE and SMSE for Burger's equation.

### Clarity on Residual Implementation
Section 2.1 mentions that finite difference methods are commonly used to approximate the differential operators, and Appendix C provides the governing PDE equations. However, the specific details on the implementation of the physics residual operator used in loss functions, such as the exact finite difference stencils. order of accuracy, handling of initial conditions/boundary conditions are missing. While readers may find the details in the code provided, could the authors include theses important implementation details directly in the appendix?

### Minor typos
- Line 351-352: Burger's equation -> Burgers' equation
- Table 1: Burger -> Burgers

---

### Official Review · Reviewer_i1HF · 2025-10-31

**Soundness:** 2
**Presentation:** 3
**Contribution:** 2
**Rating:** 4
**Confidence:** 4

**Summary:**

The paper proposes a new method for PDE-constrained generative modelling tasks. This task is important in scientific and engineering applications. The main idea is to inject the PDE constraints into a post-processing distillation stage. It views the trained diffusion model as a teacher model and uses its generated trajectory to distill a one-step generation student model under the regularization of the PDE error penalty. The method can be applied to forward, inverse, and reconstruction PDE problems. Experiments on several benchmarks demonstrate its effectiveness.

**Strengths:**

1. It proposes a new perspective to solve the challenge of PDE constraint in generation. This perspective is quite different from previous approaches that suffer from the inaccuracy caused by Jensen's gap.
2. The presentation is clear and easy to follow.
3. The experiments are convincing. The compared methods are complete and the results of the proposed method are strong.

**Weaknesses:**

1. For the contributions claimed in the Introduction Section, the "empirical confirmation of Jensen’s gap" can hardly be listed as one of the main contributions. Jensen’s gap uses $\mathbb{E}(x_0|x_t)$ to estimate clean samples from $x_t$ as a replacement for the real unknown $x_0$ in optimizing a certain objective or satisfying certain constraints. This is a widely adopted practice and recognized issue in the diffusion models community. Thus, the empirical confirmation of this gap in generative modelling of PDE domains seems not to be a significant contribution.
2. For "theoretical soundness", another listed contribution in the Introduction Section, although this method bypasses the Jensen’s gap of previous methods, it does not guarantee the hard satisfaction of PDE constraints, as it uses the norm of the residual error R(x) as a soft penalty in Eq. (8).
3. In experiments, there is no evaluation of efficiency. Thus, "efficient inference", the third claimed contribution, is not supported.

**Questions:**

Besides the questions about efficiency of the proposed listed in Weaknesses, I also have the following two questions.
1. What are the meanings of VP and sub-VP in Line 426 and Table 3?
2. The implementation of using Distribution Matching Distillation (DMD), Rectified Flow, and consistency model for one-step generation from Line 309 to Line 313 is missing. I think these details are important for high-quality one-step generation.

---

### Official Review · Reviewer_FzbT · 2025-10-31

**Soundness:** 2
**Presentation:** 2
**Contribution:** 2
**Rating:** 2
**Confidence:** 4

**Summary:**

Summary:

PIDDM is a post-hoc distillation scheme: train a vanilla teacher, generate $\varepsilon, x_0$ pairs deterministically, and fit a one-step student that penalizes the PDE residual on the final sample, with optional latent refinement. On Darcy, Poisson, Burgers, and others, it reports lower PDE residuals and competitive MMSE/SMSE/FPD, with refinement improving results. The work motivates this via a mismatch (“Jensen’s Gap”) when constraining $\mathbb{E}[x_0 \mid x_t]$ instead of samples.


Contributions:

1. Physics-informed distillation that applies a PDE residual to generated samples via a post-hoc, one-step student, aiming to decouple physics enforcement from the diffusion trajectory.

2. Unified and efficient inference: one-step generation with an optional refinement step; a single student is used for forward, inverse, and reconstruction through masked mixing and latent optimization.

3. Illustrative evidence regarding the Jensen’s Gap: examples highlighting the mismatch that can arise when constraints are imposed on $\mathbb{E}[x_0 \mid x_t]$, including a MoG toy and training-time comparisons.

4. Reported empirical results on the presented PDE benchmarks indicating lower PDE residuals and competitive generative metrics for the student, with further gains observed after refinement.

**Strengths:**

1) Originality — A post-hoc distillation that directly penalizes the PDE residual on final samples $x_0$ via $\|R(x_0)\|^2$ reduces the mismatch from enforcing $R\!\left(\mathbb{E}[x_0\mid x_t]\right)$; a single distilled model supports forward, inverse, and reconstruction using masked latent optimization $x_{\mathrm{mix}}=x'\odot M+d_{\theta'}(\epsilon)\odot(1-M)$, a creative combination that removes multi-stage guidance limits.

2) Quality — Experiments across multiple PDEs (e.g., Darcy/Poisson/Burgers) show lower residuals $\|R(x)\|$ and competitive MMSE/SMSE/FPD, with further gains after a light refinement; ablations over teacher steps $N_s$, training weight $\lambda_{\text{train}}$, and inference weight $\lambda_{\text{infer}}$ indicate stable behavior and sensible hyperparameter regimes.

3) Clarity — The paper cleanly defines $R(x)$, highlights the Jensen mismatch $R\left(\mathbb{E}[x_0\mid x_t]\right)\neq \mathbb{E}[R(x_0)\mid x_t]$, and presents succinct train/inference procedures (single-step student, masked reconstruction) with figures and tables that make the efficiency–accuracy trade-offs easy to trace.

4) Significance — It achieves lower per-sample inference cost than multi-step guidance while maintaining accuracy, and its unified distilled model extends readily to forward, inverse, and reconstruction settings—advancing practical PDE-constrained generative modeling.

**Weaknesses:**

1. No theoretical guarantee for the “Jensen’s Gap” claim: the paper asserts a mismatch $R(\mathbb{E}[x_0\mid x_t]) \neq \mathbb{E}[R(x_0)\mid x_t]$ (Eq. 4) and says the method “bypasses” it, but provides no proposition/bound/consistency theorem; the Limitations section does not add theory.

2. Navier–Stokes residual use is not ablated: the NS setup is described, but it is unclear whether the residual is enforced over full space–time versus endpoints/boundaries, and no ablation is reported to assess this choice.

3. End-to-end compute accounting is incomplete: efficiency claims do not include the cost of teacher-trajectory sampling and periodic pair refresh.

4. Results depend on the discretized residual $R$ without robustness checks: the paper itself notes reliance on a “reliable PDE residual operator,” especially when finite-difference schemes are coarse/low-accuracy; no mesh/stencil/BC sensitivity is shown.

5. Evidence for the gap during training is narrow: Fig. 2d shows an increase in the diffusion (ELBO) loss when adding a PDE term, but PDE loss and total loss are not jointly plotted, so the rise may reflect multi-objective trade-offs rather than a direct measurement of the gap.

**Questions:**

1. Could you consider adding a brief formal statement—under reasonable simplifying assumptions—clarifying how enforcing constraints on samples (rather than on $\mathbb{E}[x_0 \mid x_t]$) is expected to affect PDE residuals and distributional fidelity?

2. Would you be willing to report end-to-end compute (teacher training, teacher sampling for $\varepsilon, x_0$ pairs with refreshes, distillation, inference) in GPU-hours and wall-clock, alongside each baseline, to help readers assess overall efficiency?

3. For Navier–Stokes, could you clarify whether the PDE residual is enforced over space–time or mainly at endpoints/boundaries, and, if feasible, include an ablation (space–time vs. endpoints-only vs. boundary-only)?

4. If possible, could you share a sensitivity analysis for the supervision-pair generation (teacher step count, pair-set size, and refresh frequency) to illustrate the trade-offs?

5. Would you consider a robustness study on the discretization of $R$ (grid resolution, stencil order, boundary handling, mild operator misspecification) to understand how these choices influence outcomes?

6. Finally, would you clarify whether PDE residuals and MMSE/SMSE are reported as absolute or relative/normalized quantities, and consider adding a normalized variant to facilitate cross-dataset comparisons?

---

### Official Review · Reviewer_Fu51 · 2025-10-31

**Soundness:** 3
**Presentation:** 2
**Contribution:** 2
**Rating:** 6
**Confidence:** 4

**Summary:**

The paper introduces Physics-Informed Distillation of Diffusion Models (PIDDM), a two-stage framework that enforces PDE constraints on generated samples rather than on posterior means, addressing the *Jensen’s gap* in existing methods. It claims to achieve *both physical fidelity and generative quality* while enabling one or a few-step inference. The authors demonstrate broad applicability across forward, inverse, and reconstruction tasks in PDE generation settings.

**Strengths:**

**S1.** The paper presents a clear and well-motivated solution to the Jensen’s gap problem in physics-informed diffusion models through a principled post-hoc distillation framework that enforces constraints directly on generated samples.

**S2.** The approach is conceptually simple yet effective, showing strong and consistent performance across diverse PDE benchmarks and tasks such as forward, inverse, and reconstruction problems, outperforming competitive baselines.

**S3.** The experiments include well-designed ablations and illustrative toy studies, and the paper is clearly written and easy to follow, making the contributions accessible and convincing.

**Weaknesses:**

**W1.** The theoretical and methodological novelty of the work is incremental. While the paper presents a clear and convincing empirical illustration of the Jensen’s gap, the idea of distillation in constrained generative modeling has already been explored in frameworks such as rectified flows and consistency models. The proposed post-hoc distillation strategy largely extends existing one-step distillation approaches rather than introducing a fundamentally new formulation. It also remains unclear whether the observed gains stem primarily from the physics-informed residual loss or simply from the benefits of distillation itself. Furthermore, the stronger-performing variant, PIDDM-ref, achieves its improvements through gradient-based noise refinement that closely mirrors the optimization procedure of D-Flow, making the most effective component of the method derivative of established approaches.


**W2.** The datasets used in the paper are relatively simple and low-dimensional, which limits the strength and generalizability of the empirical conclusions. In several benchmarks, both the teacher and the ECI baseline already achieve very low PDE residuals, often near numerical precision, making the reported improvements marginal in absolute terms. The paper should clarify the dataset composition, including training and validation sizes, and evaluate whether PIDDM maintains its advantages on more challenging settings. Since the distillation formulation effectively optimizes the residual loss as a constraint, it would be valuable to test its performance on harder PDEs where optimizing PINN objectives is known to exhibit failure modes, such as those described in [1, 2].


**W3.** The paper would benefit from a more detailed computational analysis. Although PIDDM-1 and PIDDM-ref are distinguished conceptually, the work does not present explicit runtime or wall-clock comparisons. Since PIDDM-ref incorporates gradient-based refinement at inference, while other baselines rely on multi-step sampling with varying computational costs, providing wall-clock times would help clarify the true efficiency trade-offs and strengthen the empirical claims. Additionally, the related work section could be more focused; emphasizing the most directly comparable methods such as [3] and [4] would help clarify the paper’s position and distinct contributions.


**References**

[1] Krishnapriyan, Aditi, et al. *Characterizing Possible Failure Modes in Physics-Informed Neural Networks.* *Advances in Neural Information Processing Systems* 34 (2021): 26548–26560.

[2] Rathore, Pratik, et al. *Challenges in Training PINNs: A Loss Landscape Perspective.* arXiv preprint arXiv:2402.01868 (2024).

[3] Utkarsh, U., et al. *Physics-Constrained Flow Matching: Sampling Generative Models with Hard Constraints.* arXiv preprint   arXiv:2506.04171 (2025).

[4] Christopher, J. K., S. Baek, and N. Fioretto. *Constrained Synthesis with Projected Diffusion Models.* *Advances in Neural Information Processing Systems* 37 (2024): 89307–89333.

**Questions:**

**Q1**: Has the method been evaluated under out-of-distribution constraint conditions, where the enforced constraints differ from those observed in the training distribution?

**Q2.** Can the authors disentangle the contributions of the physics-informed residual loss and the distillation process itself? For instance, how does a pure distillation baseline (without residual loss) compare to PIDDM in terms of residual error and generative quality?

---

### Note · Authors · 2025-12-02

I have read and agree with the venue's withdrawal policy on behalf of myself and my co-authors.